# Spatio-temporal variations of lateral and atmospheric carbon fluxes from the Danube Delta

Marie-Sophie Maier[1,2], Cristian R. Teodoru[1], Bernhard Wehrli [1,2]

[1]Institute of Biogeochemistry and Pollutant Dynamics, ETH Zürich, Zürich, 8092, Switzerland
[2]Eawag, Swiss Federal Institute of Aquatic Science and Technology, Kastanienbaum, 6047, Switzerland

*Correspondence to*: Marie-Sophie Maier (marie-sophie.maier@usys.ethz.ch)

**Abstract**

River deltas with their mosaic of ponds, channels and seasonally inundated areas act as the last continental hotspots of carbon turnover along the land-ocean aquatic continuum. There is increasing evidence for the important role of riparian wetlands in the transformation and emission of terrestrial carbon to the atmosphere. The considerable spatial heterogeneity of river deltas, however, forms a major obstacle for quantifying carbon emissions and their seasonality: the water chemistry in the river reaches is defined by the upstream catchment, whereas delta lakes and channels are dominated by local processes such as aquatic

primary production, respiration or lateral exchange with the wetlands. In order to quantify carbon turnover and emissions in the Danube Delta, we conducted monthly field campaigns over two years at 19 sites spanning river reaches, channels and lakes. Here we report greenhouse gas fluxes ($CO_2$ and $CH_4$) from the freshwater systems of the Danube Delta and present the first seasonally resolved estimates of its freshwater carbon emissions to the atmosphere. Furthermore, we quantify the lateral carbon transport of the Danube River to the Black Sea.

We estimate the delta's $CO_2$ and $CH_4$ emissions to 65 Gg C $yr^{-1}$ (30–120 Gg C $yr^{-1}$, range calculated using 25–75 percentiles of observed fluxes), of which about 8% are released as $CH_4$. The median $CO_2$ fluxes from river branches, channels and lakes are 25, 93 and 5.8 mmol $m^{-2}$ $yr^{-1}$, respectively. Median total $CH_4$ fluxes amount to 0.42, 2.0 and 1.5 mmol $m^{-2}$ $yr^{-1}$. While lakes do have the potential to act as $CO_2$ sinks in summer, they are generally the largest emitters of $CH_4$. Small channels showed the largest range in emissions including a $CO_2$ and $CH_4$ hotspot sustained by adjacent wetlands. The channels thereby contribute

disproportionately to the delta's emissions considering their limited surface area. In terms of lateral export, we estimate the net total export (DIC+DOC+POC) from the Danube Delta to the Black Sea to about $160 \pm 280$ GgC $yr^{-1}$, which only marginally increases the carbon load from the upstream river catchment ($8490 \pm 240$ GgC $yr^{-1}$) by about 2 %. While this contribution of the delta seems small, deltaic carbon yield (45.6 gC $m^{-2}$ $yr^{-1}$, net export load/surface area) is about 4-fold higher than the riverine carbon yield from the catchment (10.6 gC $m^{-2}$ $yr^{-1}$).

## 1 Introduction

In an attempt to improve global climate models, the role of rivers, their deltas and estuaries in the carbon cycle is receiving increasing attention since more than a decade (IPCC, 2007). Back then, the perception shifted from rivers as mere lateral conduits of particulate and dissolved carbon species to an "active pipe" concept, where rivers are considered efficient biogeochemical reactors with the potential to release significant amounts of carbon as $CO_2$ and $CH_4$ directly to the atmosphere

(Cole et al., 2007; IPCC, 2013). A multitude of global upscaling studies (e.g. Tranvik et al., 2009; Regnier et al., 2013; Raymond et al., 2013) estimated the riverine and lacustrine fluxes of $CO_2$ and $CH_4$ to the atmosphere on a persistently fragmentary database considering spatial and temporal coverage, especially of headwater streams and large lowland rivers (Hartmann et al., 2019; Drake et al., 2018).

Along the land-ocean-aquatic continuum, about 0.9–0.95 PgC $yr^{-1}$ are estimated to be transferred laterally by rivers to the

ocean (Regnier et al., 2013; Kirschbaum et al., 2019). Half of the carbon exported to the ocean is in the form of dissolved

inorganic carbon (DIC), while the other half consists of particulate and dissolved organic carbon (POC and DOC) in about equal shares (Li et al., 2017; Kirschbaum et al., 2019). Recent estimates suggest that about 50 to >70 % of the carbon inputs from terrestrial ecosystems degas as $CO_2$ and $CH_4$ along the way to the ocean (Drake et al., 2018; Stumm and Morgan, 1981; Kirschbaum et al., 2019; Cole et al., 2007), making this the most important export flux of terrestrial carbon from inland waters.

While rivers could emit 0.65–1.8 PgC yr$^{-1}$ (Lauerwald et al., 2015; Raymond et al., 2013), lakes and reservoirs could add another 0.3–0.58 PgC yr$^{-1}$ (Raymond et al., 2013; Holgerson and Raymond, 2016). Earlier works on inner estuaries, salt marshes and mangroves estimate their contribution to another 0.39–0.52 PgC yr$^{-1}$ (Borges, 2005; Borges et al., 2005). So river deltas and estuaries seem to contribute about equally to $CO_2$ and $CH_4$ emissions as lakes and reservoirs, despite representing only about 1/6 of their global surface area (Cai et al., 2013; Holgerson and Raymond, 2016).


Deltas and estuaries represent hot spots of carbon turnover and $CO_2$ and $CH_4$ emissions due to high nutrient load, large productivity and seasonal flooding. However, differences in geomorphology, anthropogenic alterations, complex hydrology and the influence of tides are just a few of the factors which make it very difficult to compare different deltaic and estuarine systems amongst each other (Galloway, 1975; Postma, 1990). Dürr et al. (2011) attempted to classify this diverse group of

coastal habitats, which led to lower global emission estimates of 0.27±0.23 PgC yr$^{-1}$ of $CO_2$ and 0.0018 PgC yr$^{-1}$ of $CH_4$ (Laruelle et al., 2010; Borges and Abril, 2011). These studies, however, did not explicitly consider deltas and inner estuaries of large rivers such as the Amazon, Changjiang, Congo, Zambezi, Nile, Mississippi, Ganges or the Danube.

The close connection of river deltas to adjacent wetlands has the potential to fuel $CO_2$ and $CH_4$ emissions. Almeida et al. (2017) show that peak concentrations of $CO_2$ in the Madeira River, a tributary to the Amazon, are linked to extreme flood

events and riparian wetlands in the Amazon basin have been identified as significant sources for the outgassing of terrestrial carbon in the form of $CO_2$ (Richey et al., 2002; Mayorga et al., 2005; Abril et al., 2014). Global wetlands were estimated to contribute 1.1 PgC yr$^{-1}$ (Aufdenkampe et al., 2011) to the carbon emissions in the land-ocean aquatic continuum. The uncertainty of these estimates is large, due to the difficulty to delineate global wetland areas (Tootchi et al., 2019) and the complex interaction between potential emissions and carbon uptake by vegetation and soils (Hastie et al., 2019). While the

lower river basins of Amazon, Mississippi and Zambezi have been subject to $CO_2$ and $CH_4$ evasion studies (Sawakuchi et al., 2014; Dubois et al., 2010; Teodoru et al., 2015), others such as Nile and Danube remained unchartered territory in that respect. Both Nile and the Danube River represent one end of the river delta spectrum since they show little exposure to tidal action. Therefore, these deltas experience seasonal flooding, instead of (semi)-diurnal flooding determined by tidal action. Flooding can, in addition to groundwater drainage and surface runoff, transport substantial amounts of terrestrial carbon to aquatic

systems (Abril and Borges, 2019). We thus anticipate seasonal variability in $CO_2$ and $CH_4$ emissions and in lateral carbon transport from the Danube Delta to the ocean.

In this study, we estimate delta-scale atmospheric $CO_2$ and $CH_4$ emissions for the Danube Delta, as well as the lateral carbon transport of the Danube River to the Black Sea. We hypothesized that the hydromorphology of the different waterscapes would

influence the outgassing behavior of greenhouse gases by governing gas exchange and biogeochemical processes. The resulting differences in atmospheric fluxes would require treating the waterscapes separately in the upscaling process. Furthermore, we anticipated the seasonality of the flooding to affect both atmospheric and lateral fluxes.

To capture this spatial and temporal variability, we conducted a systematic study covering 19 sites in the Danube Delta over two years with monthly sampling intervals. Based on this time series, we address the systematic differences between the delta's main waterscapes (river branches, channels and lakes) to classify different open-water sources for greenhouse gas emissions and dominating biogeochemical processes. Furthermore, we estimate lateral and atmospheric carbon fluxes considering the spatio-temporal variability, discuss uncertainties linked to the upscaling process and compare the estimates to other major river systems.

## 2 Methods

### 2.1 The Danube Delta

The Danube Delta is the second largest river delta in Europe after the Volga Delta. It is located on the Black Sea coast in eastern Romania and southern Ukraine (Fig. 1). Close to the city of Tulcea, the Danube River splits and forms the Chilia, Sulina and Saint George branch (romanian: Sfantu Gheorghe). In the vast wetland area between the main river sections, the seasonal floods maintain an aquatic mosaic of reed stands and more than 300 shallow through-flow lakes of different sizes, which are hydrologically connected to the Danube via natural and artificial channels (Oosterberg et al., 2000). Since 1998, the Danube Delta has been a UNESCO Biosphere Reserve with nearly 10% strictly protected area and another 40% of the total surface area declared as buffer zones (UNESCO). While five of the larger lakes of the Danube Delta have been subject to $CO_2$ and $CH_4$ evasion studies in the past (Durisch-Kaiser et al., 2008; Pavel et al., 2009), the main branches of the river and the small channels are unchartered territory with respect to $CO_2$ and $CH_4$ concentrations and fluxes.

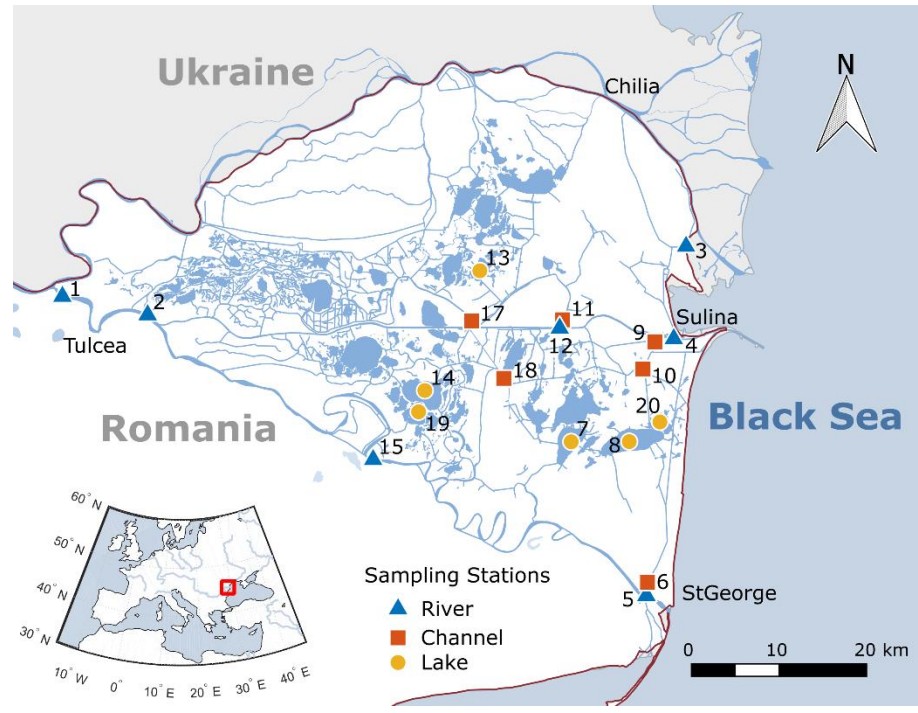


**Figure 1 Sampling stations in the Danube Delta, Romania. Near Tulcea, the Danube River splits into three branches: Chilia, Sulina and St. George. Station 16 was removed from the study because of limited access during lower water level (clogged access channel). Shape files for map creation in QGIS adapted from mapcruzin.com (Contains information from www.openstreetmap.org, which is made available here under the Open Database License (ODbL), https://opendatacommons.org/licenses/odbl/1.0/).**

*Hydrology*

The hydrology of the Danube River, which drives water exchange with the delta, has a pronounced seasonality. Receiving meltwater from the Alps and Carpathians, the Danube shows peak discharge in spring from April to June (Fig. 2), whereas the discharge minimum occurs in autumn from September through November. December and January often show a small peak in discharge. The discharge provided by the Danube River drives the seasonal and annual hydrological changes in the delta. From

2000 to 2014, the Danube's average annual discharge was 6760 $m^3\ s^{-1}$ (ICPDR, 2018), which is a 3 % increase compared to the period from 1930 to 2000 (Oosterberg et al., 2000). In the delta region, the discharge splits into the different main branches as follows: Chilia: 53 %, Sulina: 27 %, Saint George branch: 20 % (ICPDR, 2018). Approximately 10 % of the Danube's total discharge (620 $m^3\ s^{-1}$, averaged 1981-1990) flows through the delta, of which about 20 % (120 $m^3\ s^{-1}$) is lost via evapotranspiration (Oosterberg et al., 2000).

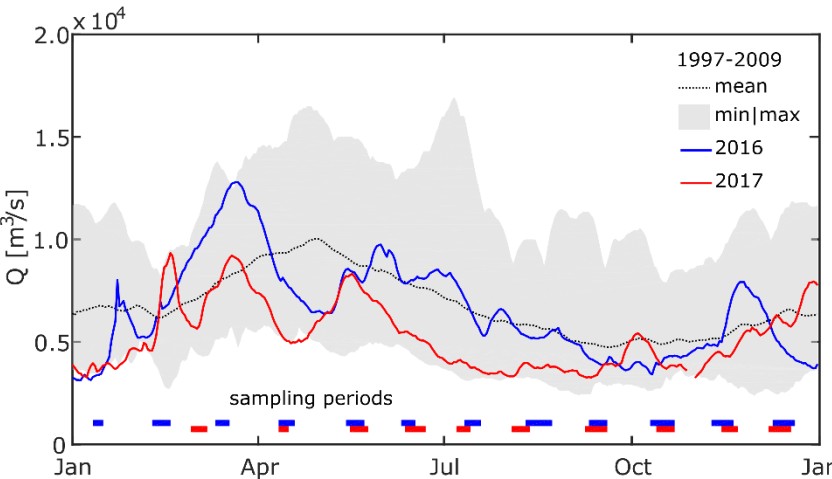

**Figure 2 Daily average discharge close to the apex of the Danube Delta. The dotted line and the shaded area show mean and minimum to maximum daily discharge, respectively, for the period from January 1997 to October 2009 at Reni (ICPDR, 2018). Blue and red lines show daily average discharge at Isaccea in 2016 and 2017 (INHGA; Feodorov). Horizontal bars indicate the timing of sampling campaigns. X-axis ticks indicate date 15 of the respective month.**

To assess the hydrological conditions during the time of observation with respect to the long-term average, we compared water

level observations from Isaccea (INHGA; Feodorov, 2017) to the discharge data set from Reni (ICPDR, 2018). Reni is located about 30 km upstream of Isaccea, without any major tributary joining in between. Water level data from Isaccea was converted to discharge using rating curves created from paired water level and discharge data from the National Institute of Hydrology and Water Management (INHGA). The comparison shows that 2016 was quite an average year in terms of discharge (Fig. 2), while, contrastingly, the Danube had very low discharge in 2017, especially during the period between March and October.

Average discharge in 2017 was 5237 $m^3$ $s^{-1}$, or 23 % below the average flow calculated from the ICPDR data set, hence we refer to it as "dry year". Water temperature and conductivity of our sampling period were also in general comparable with data from the ICPDR's long-term monitoring (see SI). Although water temperature measured during summer months in both 2016 and 2017 was up to 3°C warmer than the long-term mean, these values did not exceed maximum temperatures measured in the last 20 years.

*Categorization into river branches, channels and lakes*

We categorized our sampling stations into three groups based on geomorphological characteristics: main river branches, lakes and channels. River branch stations are all located along the three main branches of the Danube River exhibiting velocities of about 0.75 m $s^{-1}$ (Danube Commission, 2018), large hydraulic cross-section and frequent embankment. The category lake refers to shallow (2-3.5 m) open-water bodies within reed bed areas, and 5 out of 6 sampling stations showed abundant

macrophytes in summer. Natural and artificial channels represent the third category. They provide a surface water connection between the lakes and the river branches. We included old meanders of the Danube as well as small channels within the delta. Both of these features show a low flow velocity of up to 0.3 m $s^{-1}$, yet span quite a range in terms of surface area and depth.

Accessibility by motor boat determined the sampling stations in lakes and channels and restricted our monitoring to deeper lakes and larger channels. Both lakes and channels are connected to adjacent reed beds and marsh areas. Very shallow or isolated lakes, which are not represented in our data set, may receive a significant part of their water from adjacent reed beds (Coops et al., 2008) and have a higher residence time of up to 300 days compared to the investigated lakes, which have an estimated residence time of 10 – 30 days (Oosterberg et al., 2000).

## 2.2 Sampling

Our research area was located in the southern part of the delta enclosed by the Sulina and Saint George branches, which we studied intensively in 2016 and 2017. We focused on the southern part of the Delta, since it is less impacted by agriculture compared to the area north of the Sulina branch (Niculescu et al., 2017). Samples and in-situ measurements were taken once per month at 19 stations (Fig. 1), representing river main branches (n = 7), channels (n = 6) and the larger delta lakes (n = 6). The sampling stations in the channels and lakes cover both the fluvial (west of station 18, Fig. 1) and the fluvio-marine parts of the delta. In-situ measurements and sampling with a Niskin bottle was carried out 50 cm below the water surface. Sample analyses were conducted at the Eawag laboratories in Switzerland.

## 2.3 Dissolved and particulate carbon species

Samples for DIC measurements were filtered sterile (0.2 µm) and bubble-free into 12 mL Labco Exetainers and stored cool and dark until analysis using a Shimadzu TOC L Analyzer. For the analysis of POC and DOC, water was filtered through 7 µm pre-combusted and pre-weighed Hahnemühle GF 55 filters. The filters were stored at -20°C until analysis, when they were dried and weighed for total suspended matter, subsequently fumigated with HCl for 24 hours to remove the inorganic fraction and analyzed by EA-IRMS (elemental analyzer) for organic carbon content, which we used to calculate POC. The filtered water was acidified using 100 µL 10 M HCl and stored dark at 4°C until analysis of DOC using a TOC L Analyzer (Shimadzu). Due to potential contamination during sampling, DOC data prior to May 2016 was discarded.

## 2.4 Dissolved gases

### 2.4.1 Concentration measurements

We used mostly field-based methods for the analysis of dissolved $CH_4$, $CO_2$ and $O_2$. In 2016, samples for $CH_4$ analysis were taken for laboratory-based analysis by gas chromatography. Water was filled bubble-free into 120 mL septa vials by allowing overflow of approximately 3 times the sample volume before preserving the sample by adding $CuCl_2$. Depending on the expected concentrations, a headspace of 15-25 mL was created in the lab using pure $N_2$. Samples were equilibrated overnight at 23°C on a shaker and the headspace was analyzed using gas chromatography (GC-FID, Agilent Technologies, US). In 2017, we used 1 L Schott-Bottles to prepare headspace equilibration directly in the field using air. Samples were transferred to gasbags and analyzed in the field for $CH_4$ using an Ultraportable $CH_4$/$N_2O$ analyzer (Los Gatos Research, LGR). We corrected

for atmospheric contamination during the processing by subtracting the amount of $CH_4$ introduced with the air during equilibration. As tests showed that there was no significant difference between the lab- and field-based methods (see SI), we pooled the data in our analysis. $CO_2$ concentrations were measured in the field using syringe-headspace equilibration of 30 mL sampling water with 30 mL air. The syringes were shaken for 2 minutes and allowed to equilibrate before transfer of the headspace into a dry syringe and analysis in an infrared gas analyzer (EGM-4, PP-Systems). The method is explained in more detail in Teodoru et al. (2015).

Dissolved $O_2$ concentration was measured in-situ using an YSI ODO probe. The sensor was calibrated daily using water-saturated air and cross-checked with oxygen readings from an YSI PROPlus multimeter sensor. We measured local in-stream respiration rates to evaluate if community respiration could sustain our measured $CO_2$ fluxes. The respiration rate was measured as $O_2$ drawdown over a 24-hour period. For the measurement, six BOD bottles were filled with water sample and three were measured immediately afterwards at t = 0. The other three bottles were stored in the dark at approximately in-situ temperatures and $O_2$ concentration was measured after 24 hours. The $O_2$ consumption rate was derived from the time and concentration difference, assuming a linear decrease over time. We used this respiration rate to estimate local $CO_2$ production rate by assuming a 1:1 aerobic respiration relation of $O_2$:$CO_2$. Ward et al. (2018) argue that respiration rate measurements in BOD bottles underestimate respiration rate because microbial processes are limited by both the bottle size and the lack of turbulence and suggest a correction factor of 2.7 to correct BOD derived respiration rates for size effects only or a factor of 3.7 for size and low turbulence effects. Applying these correction factors did not change the main point of our comparison between fluxes and $CO_2$ production rates.

### 2.4.2 $CO_2$ and $CH_4$ flux measurements

$CO_2$ and $CH_4$ fluxes were measured using a floating chamber. The chamber had an internal area of 829.6 $cm^2$ and an internal volume of 10080 $cm^3$, leading to a Volume/Area ratio of 12.15 cm. An aluminum foil coating minimized heating during deployment. $CO_2$ was routinely measured in the field over a 30-minute period by coupling an infrared gas analyzer (EGM-4, PP Systems) to the chamber in a closed loop. In 2016, $CH_4$ was sampled from the chamber by syringe and transferred overhead into 60 mL septa vials that had been pre-filled with saturated NaCl solution until the liquid was replaced by gaseous sample. These discrete samples for lab analysis were taken at time t = 0, 10, 20 and 30 min and analyzed by GC-FID. In 2017, this laborious procedure was replaced by attaching the LGR directly to the floating chamber.

Flux chamber measurements were conducted unless conditions were too windy or boat traffic was too frequent in the main channel. In total, we took 265 flux measurements for $CO_2$ and 122 for $CH_4$. Of the latter, 91 measurements seemed to be without significant influence of ebullition (i.e. $R^2$ of linear regression > 0.96, for more detail see SI) and are henceforth referred to as *diffusive* $CH_4$ fluxes. In the high-resolution LGR time series, the influence of gas bubbles could easily be identified. We calculated the diffusive flux by fitting a linear regression to periods where data showed no influence of ebullition. In this case, the flux is calculated from the slope and the height of the gas volume in the chamber. In the discrete time series, it was hard to distinguish between diffusive flux and ebullition. When the linear regression of the discretely measured samples had an $R^2 <$

0.96, we considered the flux measurement to be influenced by bubbles. In this case, we calculated the total flux by dividing the total concentration increase by the observation time, as we did to calculate the total flux of the LGR measurements. Three cases with $R^2 > 0.96$ showed fluxes $> 20$ mmol m$^{-2}$ d$^{-1}$ and were thus also classified as total flux. Discrete time series showing a non-monotonous course (n = 12) were excluded from further processing. Missing monotony can have several explanations including sampling captured a bubble or a sample mix up.

*Calculation of $k_{600}$*

We used our $CO_2$ flux measurements to calculate the gas transfer coefficient $k_{600}$ as follows

$$k_{CO_2} = \frac{F_{CO_2}}{(p_{CO_2,water} - p_{CO_2,air}) \cdot K_{H,CO_2}} \tag{1}$$

$$k_{600} = \frac{k_{CO_2}}{(Sc_{CO_2}/600)^{-\frac{1}{2}}} \tag{2}$$

Where $F_{CO_2}$ is the flux of $CO_2$, $p_{CO_2}$ is the measured partial pressure of $CO_2$ in water and air, respectively, and $K_{H,CO_2}$ is the Bunsen coefficient for $CO_2$ according to Weiss (1974). $Sc_{CO_2}$ is the Schmidt number for $CO_2$ calculated based on temperature (Wanninkhof, 1992). We estimated missing flux measurements using the median $k_{600}$ of the respective water type and the measured $CO_2$ concentrations.

Analogously, diffusive $CH_4$ fluxes were estimated from the individually calculated $k_{600}$ using the Bunsen coefficient from Wiesenburg and Guinasso Jr (1979), the mean global atmospheric $CH_4$ mole fraction of 1.84 ppm (Nisbet et al., 2019) and the Schmidt number for $CH_4$ from Wanninkhof (1992). We attributed the difference between this estimate and the total measured flux to ebullition.

## 2.5 Upscaling atmospheric fluxes to delta-scale

Spatial upscaling of heterogeneous and scarce data is very difficult and handled in various ways in the literature. Like other authors in a global context (Aufdenkampe et al., 2011; Raymond et al., 2013), we believe that median fluxes give a more reliable representation of the fluxes in systems with large gradients. Based on the different characteristics of the three waterscapes, we estimated the delta-scale atmospheric $CO_2$ and $CH_4$ fluxes by multiplying the median flux of each waterscape with its respective area (Table 1). We did this for each month separately and summed up the results considering the respective number of days per month. For example, the median annual flux from the rivers, $\bar{F}_R$ was calculated as

$$\bar{F}_R = \sum_{m=1}^{12} F_{R,m} \cdot A_R \cdot n_m \cdot 10^3 \tag{3}$$

where $F_{R,m}$ is the median flux in mmol m$^{-2}$ d$^{-1}$ measured in the river stations in month $m$, $A_R$ is the area of the river branches in km$^2$ (see Table 1) and $n_m$ represents the number of days in the respective month $m$. The factor $10^3$ is used to convert to the units of mol yr$^{-1}$. To obtain the annual flux from the channels, $\bar{F}_C$, and the lakes, $\bar{F}_L$, we proceeded the same way. We converted the resulting annual fluxes of the different waterscapes from mol yr$^{-1}$ to GgC yr$^{-1}$ and GgCO$_2$eq yr$^{-1}$, the latter assuming a

global warming potential for CH$_4$ of 28 over 100 years, i.e. neglecting climate feedbacks (IPCC, 2013). The total annual water-air flux, $\bar{F}_{tot}$, from the delta was the sum of the three fluxes:

$$\bar{F}_{tot} = \bar{F}_R + \bar{F}_C + \bar{F}_L \tag{4}$$

We also performed this calculation using 25- and 75-percentiles instead of the median to assess upper and lower boundaries of our estimate.

For a reliable upscaling of fluxes, we determined the surface area of each waterscape as precisely as possible (Table 1). We estimated the area covered by the Danube's branches by refining publicly available shape files for Romania and Ukraine (mapcruzin.com, 2016) using the "Open layers plugin" in QGIS, which allowed comparison of the shape file with satellite images. We used the same procedure for the lakes and arrived at the surface area reported by Oosterberg et al. (2000). Assessment of the surface area of the delta channels was more difficult, as many of the small channels are hard to identify on satellite images. Generally, estimating the width of the channels is challenging due to emergent macrophyte coverage, which depending on image quality blends in with adjacent reed. Instead of mapping the channels, we therefore used the overall channel length reported by Oosterberg et al. (2000) and assumed an average channel width of 19 m, which means the resulting surface area is on the lower end. Especially the old, cut-off meanders of the Danube River (Dunarea Veche), which we also consider as belonging to the channel category, do have a much larger width ranging in the order of 100–200 m.

**Table 1 Surface area of Danube Delta features. Assuming 19 m channel width means the estimation of the surface area of the channels is on the lower end. The surface areas of freshwater and wetland do not add up to the total area, since parts of the delta are covered by forest and agricultural polders.**

| Feature | Area [km$^2$] | Source |
|---|---|---|
| Freshwater | 455 | Sum of river branches, channels and lakes |
| - River branches | 164 | Extracted using QGIS[*] |
| - Channels | 33 | Length of canals from Oosterberg et al. (2000); 19 m width assumed. |
| - Lakes | 258 | Oosterberg et al. (2000) & extracted using QGIS[*] |
| Wetland | 3670 | Mihailescu (2006) |
| - Marsh vegetation (total) | 1805 | Sarbu (2006) |
| o Scripo-Phragmitetum | 1600 | Sarbu (2006) |
| Agriculture, forest, settlements, pastures, fish ponds | 1515 | Total surface area – wetland - freshwater |
| Total surface area within the 3 main branches | 3510 | Niculescu et al. (2017) |

| Total surface area of the delta | 5640 | Mihailescu (2006) |
| Surface area of the Danube River catchment | 817000 | Tudorancea and Tudorancea (2006) |

[*] based on shape files adapted from mapcruzin.com (2016), contains information from www.openstreetmap.org, which is made available here under the Open Database License (ODbL), https://opendatacommons.org/licenses/odbl/1.0/).

## 2.6 Import by Danube River and Export to Black Sea

To compare the delta's $CO_2$ and $CH_4$ emissions to the lateral transfer of carbon from the catchment to the Black Sea and the influence of the delta region, we also calculated the loads of dissolved and particulate carbon species transported by the Danube River at the delta apex, $F_D$, and close to the Black Sea, $F_{BS}$. As a first step we calculated the daily average load of each month, $F_m$ for the different carbon species:

$$F_m = C_m \cdot \overline{Q_m} \tag{5}$$

where $C_m$ is the concentration of DIC, DOC or POC measured in month $m$ and $\overline{Q_m}$ is the respective averaged daily discharge of month $m$. Since $CH_4$ showed much smaller concentrations (~factor 100–1000 with respect to DOC and DIC), we did not include it in our calculation. In a second step, we weighed $F_m$ by the number of days per month $n_{days,m}$ and took the sum over all months of the year. The load transported by the Danube River upstream of the delta, $F_D$, was calculated based on the concentrations measured at station 1 (Fig. 1), which is located in the Tulcea branch close to the apex of the delta and represents the water signature from the catchment:

$$F_D = \sum_{m=1}^{12} F_m(\text{st. 1}) \cdot n_{days,m} \tag{6}$$

Data from the stations in the three main branches close to the Black Sea (station 3, 4 & 5, Fig. 1) were used to estimate the amount of carbon exported to the Black Sea, $F_{BS}$:

$$F_{BS} = \sum_{m=1}^{12} [F_m(\text{st. 3}) + F_m(\text{st. 4}) + F_m(\text{st. 5})] \cdot n_{days,m} \tag{7}$$

Stations 4 and 5 are located shortly upstream of the settlements of Sulina and St.George to avoid measuring the effect of these two settlements. Station 3 is located in a small side arm of the Chilia branch marking the border between Romania and Ukraine, which during comparison measurements showed the same water composition as the main branch.

In our data processing, we decided to exclude one unusually high POC value in April at the Sulina branch (station 4) from our load calculation as we assume it is caused by a high discharge, high turbidity event that does not represent the monthly mean well. Instead, we interpolated between March and May. For DOC, we replaced missing data from January to April 2016 by the measurements at the same stations in 2017, assuming that they are also good estimates for the previous year. This way, we arrived at DOC estimates that cover the same period as DIC and POC.

We calculated the lateral transfer of carbon between the Danube Delta and its River by subtracting the load exported to the Black Sea via the three main branches, $F_{BS}$, from the load imported to the delta from the catchment, $F_D$:

$$F_{lateral} = F_{\mathrm{D}} - F_{BS}$$

The resulting lateral flux in our case is comparably small and we used gaussian error propagation to estimate its range. The basis for the error propagation were the measurement uncertainties in concentrations (0.5% DIC, 4% DOC, 10% POC) and discharge (3%, assumed), which were used to calculate the loads.

## 2.7 Statistical analysis

We used Matlab R2016a and R2017b for the statistical analysis of the data set. The data was evaluated for normal distribution using histograms and quantile-quantile-plots. In case of $O_{2,sat}$, $CH_4$ and POC, data distribution improved towards normality using log-transformation, however the results were not fully satisfying. Levene's test furthermore revealed the heteroscedastic nature of our data. Results for tests of significant difference between the three aquatic categories from the non-parametric Kruskal-Wallis test (De Muth, 2014) followed by a multiple comparison test after Dunn-Sidak were therefore taken very cautiously. Given the non-normality of the data, we report median instead of mean values and give ranges as 25 to 75 percentiles or minimum to maximum measured value as indicated.

Boxplots shown in this paper are indicating the 25 and 75 percentiles, as well as the median. Outliers are detected using the Interquartile range (1.5* IQR). The whiskers are indicating the minimum and maximum values that are not detected as outliers by this procedure.

## 3 Results

### 3.1 Dissolved and particulate carbon species

DIC concentrations measured during our study ranged from 1.6 to 4.2 mM (Fig. 3a). Median DIC concentrations were around 3.0 mM over the whole observation period, with channels showing 10 % higher and lakes showing 3 % lower median concentrations than the main river. In 2016, concentrations were lowest in August and highest in December in all three groups (Fig. 3b). In 2017, median concentrations were 10 % (rivers) to 20 % (channels, lakes) lower than in 2016.

DOC levels in the delta were about 1.8-times the concentrations observed in the river (Fig. 3c & 3d). Channels and lakes had very similar concentrations and both showed a general increasing trend from May to October 2016 but in the river, concentrations already peaked in July 2016 and were lowest in October. Median concentrations were quite comparable for 2017, with a tendency towards lower values: DOC in the main river in August 2017 was nearly 30 % lower than in the previous year. Most of the year, DOC concentrations were nearly a factor 10 smaller than measured DIC concentrations.

In 2016, we observed the lowest median POC concentration in the channels (Fig. 3e & 3f). Median concentrations in both rivers and lakes were nearly twice as high compared to channels, but showed a distinctly different seasonality: POC was

highest in the main river from March to June, while it peaked in lakes during August to October suggesting different carbon sources.

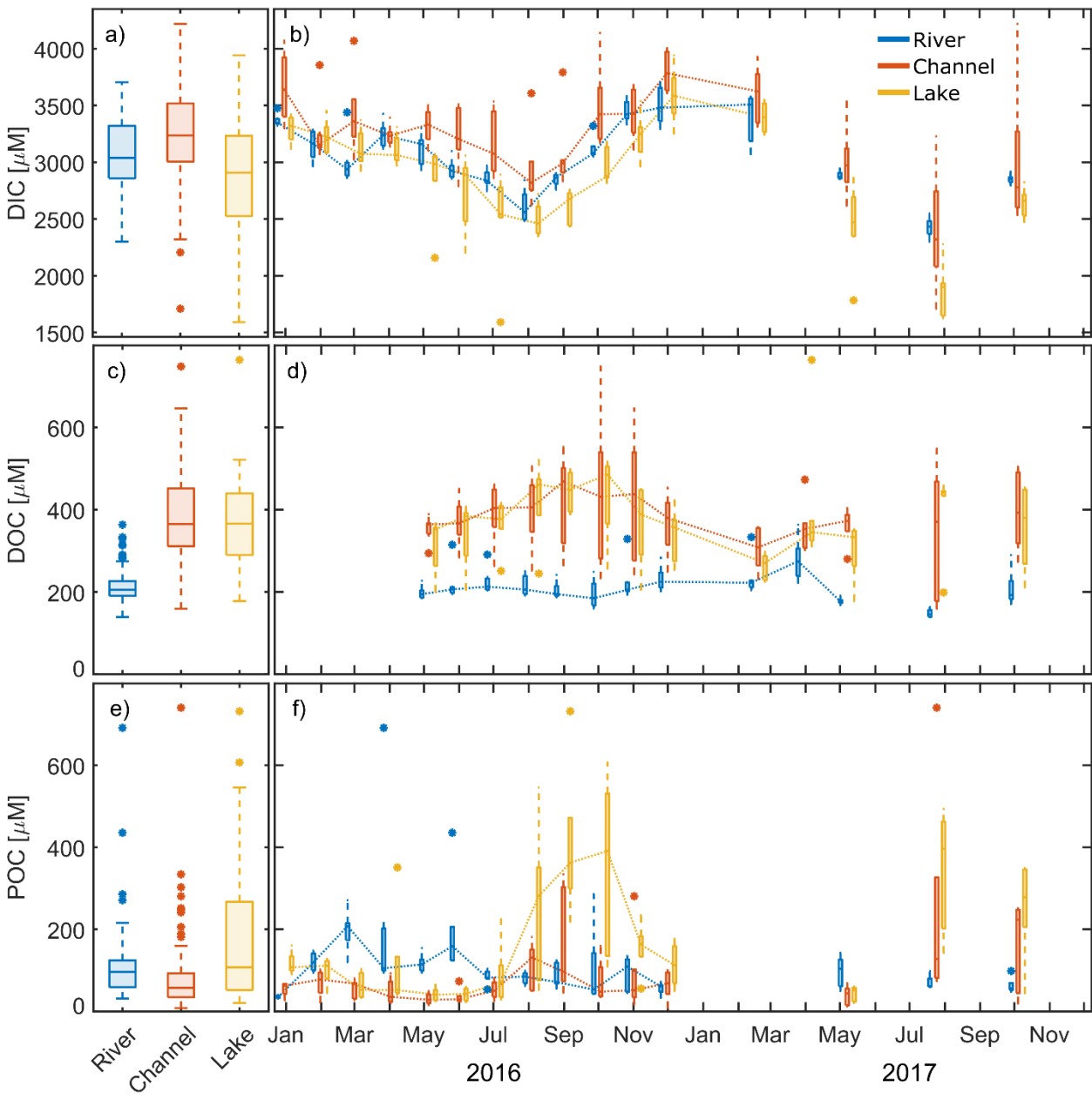

**Figure 3 Measured DIC, DOC and POC concentrations in the different waterscapes (river, channel, lake). Left panels (a, c, e): 2-years observation period. Right panel (b, d, f): seasonality of the data. Dotted lines connect median values. X-axis ticks indicate day**

## 3.2 Dissolved gases

### 3.2.1 Concentrations

During the entire monitoring period, $CH_4$ in water samples of the delta were always oversaturated with respect to atmospheric equilibrium concentrations of 0.0046 to 0.0023 µM at T = 0 to 30 °C (Fig. 4c & 4d). Median concentrations in the river samples were thus ~100 times oversaturated (0.33 µM). The channels exhibited a more than 3-times higher median concentration than the main river (1.1 µM) with highest concentrations in July to September 2016 (up to 59 µM). By contrast, the median

concentration in the lakes exceeded the value of the main river only slightly (0.43 µM), yet with a much larger range. In all three subsystems, concentrations increased from February 2016 to maximum values in July to October 2016. In 2017, concentrations were lower in the channels compared to 2016.

In analogy to $CH_4$, we found $CO_2$ concentrations to be constantly supersaturated with respect to the atmosphere in the main branches of the Danube, ranging from 26 to 140 µM (Fig. 4e & 4f). The median concentration of 59 µM was more than 3-

320 times as high as the equilibrium concentration of $CO_2$ at 15°C (18.2 µM). Channels showed a much higher range (2.4 to 790 µM) with a significantly higher median of 140 µM. During the entire monitored period, we encountered undersaturated conditions in this class at only two stations (17 and 18) in August 2017. Lakes, however, were undersaturated at 11 occasions in 2016 and 32 occasions in 2017. Dissolved concentrations in this category ranged from 0 to 95 µM with a median of 28 µM. In 2016, $CO_2$ concentration showed a pronounced seasonality in all three subsystems. In the main river, median $CO_2$ nearly

325 doubled from January 2016 to April 2016 and subsequently decreased to reach levels around 60 µM. In 2017, no clear seasonal pattern emerged. That year, median values mostly ranged around 60 µM, with lowest median concentration recorded in June (44 µM) followed by the maximum in July (81 µM).

Channels showed the largest increase of $CO_2$ during the warm season: median concentrations increased more than 4-fold, from 66 µM in February 2016 to 290 µM in July 2016. In terms of inter-annual $CO_2$ variability, 2017 showed a later and less

pronounced increase in concentration (72 µM in March to 187 µM in May) followed by an earlier decline than 2016. From August 2017 to November 2017, median monthly concentrations ranged around 50 µM and were lower than the concentrations in the main river during this period. In general, $CO_2$ concentrations in the channels in 2017 were 18 to 75 % below the values observed in 2016. We found the highest concentrations in the eastern part of the delta (station 10, Fig. 1), where concentrations reached around 360 µM in winter and up to 785 µM in summer 2016.

Compared to rivers and channels, lakes generally had the lowest $CO_2$ concentrations and showed a distinctly different seasonal pattern. Most of the observed lakes (station 7, 8, 13 and 14) were undersaturated in the period from May to November 2016. $CO_2$ undersaturation in these lakes (incl. station 20) occurred 3-times more often and over a longer period from March to

December in the dryer year 2017. In 2016, lakes showed highest median $CO_2$ concentrations in April (74.4 µM) and lowest concentrations in July and August (20.5, and 14.6 µM, respectively). With the concentration increase in early spring, the decrease in summer and the following increase in autumn, the seasonal signal in 2016 recalls a sinusoidal curve. The pattern in the drier year, 2017, however, showed less variation with lower concentrations, which were ranging from 0 to 71 µM.

$O_2$ saturation, as one might expect, often showed a mirror image to the $CO_2$ time series in all three systems (Fig. 4g & 4h). The main river was generally slightly undersaturated with a median $O_2$ saturation of 93 %. $O_2$ saturation in river water ranged between 75 and 109 % during the whole observation period. Median saturation in the channels was 14 % lower (79.5 %) and – as for $CO_2$ – covered a much broader range than in the main river: lowest values observed were as low as 5 % $O_2$ saturation (0.4 mg $L^{-1}$) in July 2016, while maximum saturation reached nearly 150 % in August 2017. In winter, $O_2$ saturation in the channels was comparable with the river stations. Station 10 showed an exceptional behavior and never exceeded a saturation of 72 % or 9 mg $L^{-1}$. $O_2$ saturation in the channels strongly decreased in spring and summer months resulting in concentrations of less than 2 mg $L^{-1}$ at stations 9 in July 2016 and at station 10 from July to September 2016 and in June, July and October 2017. Contrastingly, most lakes showed a strong oversaturation of up to 180 % from April to October, resulting in a median saturation that slightly exceeded 100 %.

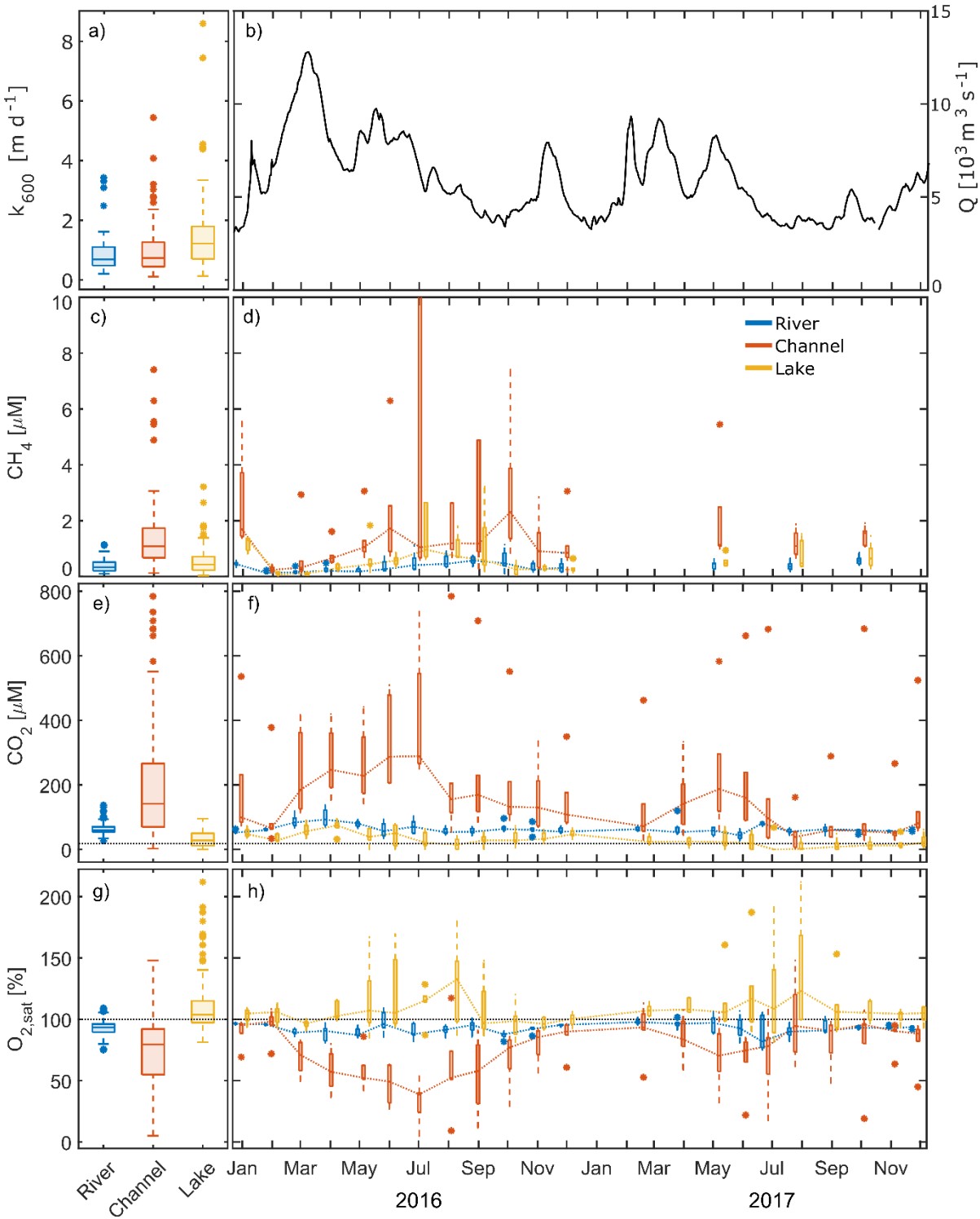

**Figure 4** $k_{600}$ (a), daily average discharge close to the apex (b) and measured concentrations of dissolved gases in the different waterscapes, i.e. river, channel and lake (c-h). Left panels (a, c, e, g): pooled data from 2-years. Right panel (d, f, h): seasonal dynamics with dotted lines connecting median values. X-axis ticks indicate day 15 of the respective month. c) & d) $CH_4$ in 2016: four channel values (ranging from 22.2 to 58.0 µM) and one lake station (12.5 µM) exceeding 10 µM were cutoff. e) & f) dotted black line represents equilibrium concentration of $CO_2$ at 15°C (18.2 µM). Boxplots indicate 25 and 75 percentiles, as well as median, whiskers indicate maximum and minimum, with data > 1.5*IQR is shown as outliers.

### 3.2.2 Measured atmospheric $CO_2$ and $CH_4$ fluxes

Median $CO_2$ fluxes were largest in channels (93 mmol m$^{-2}$ d$^{-1}$, see Table 2), where we also observed the highest overall flux of 880 mmol m$^{-2}$ d$^{-1}$. Lakes were the only locations that showed significant negative fluxes, i.e. $CO_2$ uptake during summer, when $O_2$ was strongly oversaturated.

The highest median diffusive fluxes of $CH_4$ were observed in the channels with 1.1 mmol m$^{-2}$ d$^{-1}$. Diffusive efflux from the river was generally lowest, while the lakes showed the largest variability with a minimum of 0.03 and a maximum of 6.7 mmol m$^{-2}$ d$^{-1}$. Considerable ebullition occurred only in the delta lakes and channels, which accounted for ~70% of the total $CH_4$ flux.

The gas transfer coefficient, $k_{600}$, was calculated from the measured $CO_2$ fluxes. Median $k_{600}$ was lowest in the river branches and in the channels at 0.69 m d$^{-1}$ and 0.74 m d$^{-1}$, respectively (see Table 2). As lakes were more exposed to wind, median $k_{600}$ was considerably higher (1.2 m d$^{-1}$) and we observed the maximum $k_{600}$ of 8.6 m d$^{-1}$ in this category.

**Table 2** Median and range of measured $CO_2$ and $CH_4$ fluxes [mmol m$^{-2}$ d$^{-1}$] and calculated $k_{600}$ values [m d$^{-1}$]. n states the number of measurements. The range indicates minimum and maximum observations.

| Parameter | River | | | Channel | | | Lake | | |
|---|---|---|---|---|---|---|---|---|---|
| | median | range | n | median | range | n | median | range | n |
| $F_{CO2}$ | 25 | 7.3–150 | 57 | 93 | -9.7–880 | 105 | 5.8 | -110–160 | 103 |
| $F_{CH4, tot}$ | 0.42 | 0.056–2.7 | 21 | 2.0 | 0.062–51 | 47 | 1.5 | 0.031–47 | 54 |
| $F_{CH4, dif}$[a] | 0.37 | 0.056–2.7 | 17 | 1.1 | 0.16–6.2 | 34 | 0.82 | 0.031–6.7 | 40 |
| $k_{600}$[b] | 0.69 | 0.20–3.4 | 57 | 0.74 | 0.11–5.4 | 103 | 1.2 | 0.13–8.6 | 96 |

[a] The data in this table relies only on measured $F_{CH4,dif}$. Missing diffusive $CH_4$ fluxes for the upscaling were calculated from $k_{600}$.

[b] Measurement uncertainty lead to negative $k_{600}$ values in 9 cases (n_channel = 2, n_lakes = 7). These values were deleted manually, thus n_$k_{600}$ is < n_$F_{CO2}$ for channels and lakes.

### 3.2.3 $CO_2$ production rate vs. $CO_2$ flux

We find respiration rates ranging between 0.8–390 mM m$^{-2}$ d$^{-1}$ for rivers, while in the channels and lakes they ranged from 2.3–560 mM m$^{-2}$ d$^{-1}$ and 1.0–350 mM m$^{-2}$ d$^{-1}$, respectively (Fig. 5 and Fig. S7-S9). Median respiration rate is highest in rivers

(54 mM m$^{-2}$ d$^{-1}$), followed by lakes (48 mM m$^{-2}$ d$^{-1}$) and channels (45 mM m$^{-2}$ d$^{-1}$). Many stations showed a pronounced seasonality with highest respiration rates occurring mostly between July to October. Respiration rates, i.e. $CO_2$ production rates generally exceed $CO_2$ fluxes in river and lake stations throughout the year (Fig. 5), which implies that local instream $CO_2$ production sustained the observed fluxes. At the channel stations we frequently observed fluxes exceeding the local production, even if we account for potential underestimation of the $CO_2$ production, which implies the presence of other $CO_2$ sources. This

was most striking at station 10, the $CO_2$ hotspot, where $CO_2$ outgassing exceeded local respiration on average by a factor of 40. At the other channel stations (also see Fig S8), there seems to be a seasonally occurring pattern: $CO_2$ fluxes exceed local production in the first half of the year, while for the remainder of the year they fall below. While this pattern is very distinct in 2016, it is less pronounced in the drier year 2017, which suggests that the additional $CO_2$ source is linked to hydrology.

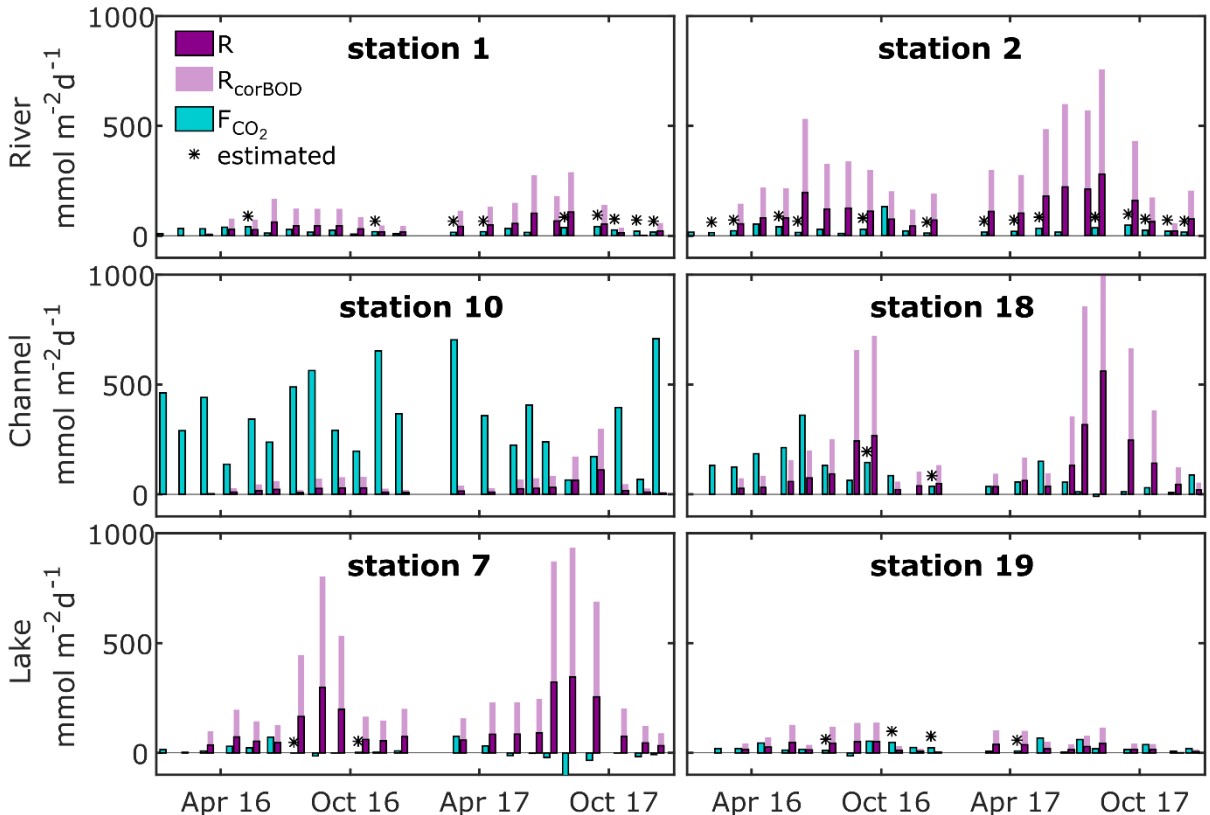

**Figure 5 Flux rate and production rate of $CO_2$ as calculated from $O_2$ community respiration incubations for selected river, channel and lake stations. Fluxes marked with asterisks were calculated from median $k_{600}$ of our observations in the respective waterscape. Dark purple bars represent measured respiration rates, light purple bars indicate the effect of a correction for measurement limitations using BOD bottles (factor 2.7, see Ward et al. (2018)).**

**3.3 Upscaling atmospheric fluxes to delta-scale**

The upscaling of the freshwater $CO_2$ and $CH_4$ fluxes to the freshwater surface of the delta according to Eq. (4) led to a net $CO_2$ flux of 60 GgC in 2016 and less than half (23 GgC) in the drier year 2017 (Fig. 6 and Fig. 7a, case "c"), when the overall contribution of the three compartments was lower and lakes turned into a net sink. The diffusive $CH_4$ flux (Fig. 7c) was one order of magnitude smaller than the $CO_2$ flux (Fig. 7a) but it increased 3-fold when ebullition was considered (Fig. 7d). Especially the $CO_2$ fluxes seem to be subject to considerable inter-annual variability (Fig. 6a & 6b), which highlights the need

to discriminate between different years during the upscaling process. It is likely that the different hydrological conditions triggered different amounts of lateral inflow from the reed-covered wetlands and contributed to the large variability in $CO_2$ fluxes. For $CH_4$, this effect appears to be much smaller.

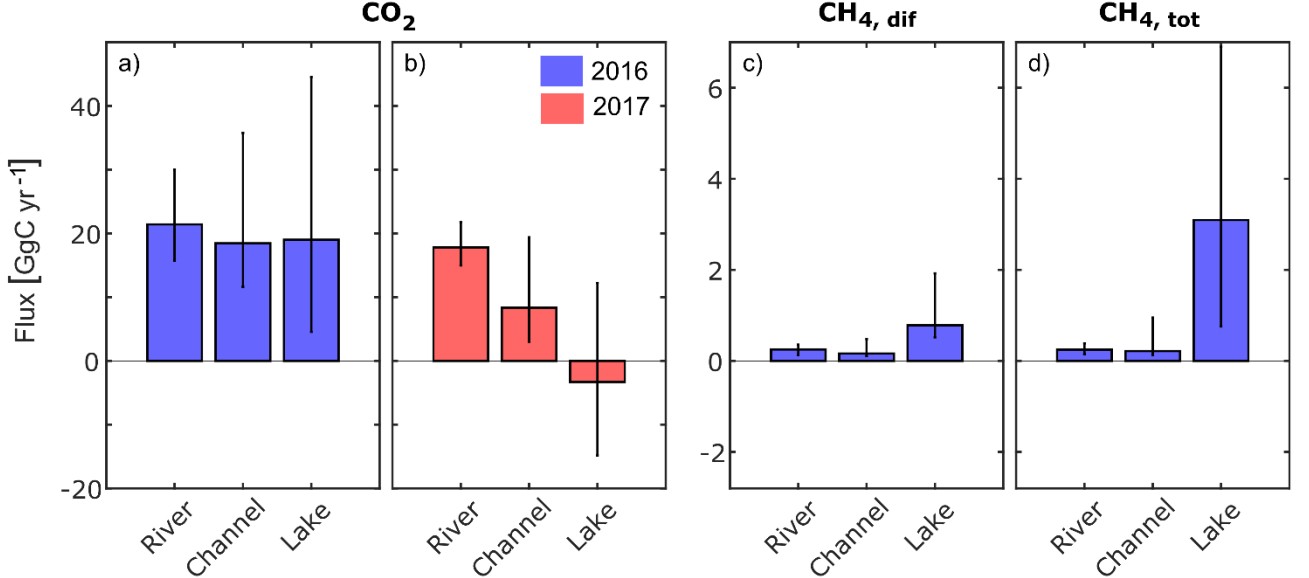

**Figure 6 Annual greenhouse gas fluxes to the atmosphere obtained by upscaling the monthly median flux to the total area of each**
**waterscape and taking the sum over all months (details see text). Black vertical lines indicate the uncertainty and were calculated by using the 25 percentile and 75 percentile, respectively, instead of median values for the calculation. $CO_2$ flux in a) 2016 and b) 2017, c) diffusive and d) total $CH_4$ flux in 2016. Due to large data gaps, this calculation was not done for $CH_4$ in 2017. All Fluxes are in GgC yr$^{-1}$. For tabulated values see Table S2.**

Considering the contributions from the different waterscapes shows that the river branches were the main source of $CO_2$ to the atmosphere in both years (Fig. 6a & 6b). Despite their small surface area (7 %), channels contributed 32–37% to the total $CO_2$ flux. Lakes on the other hand switched from a net $CO_2$ source of 19 GgC in 2016 to a small net $CO_2$ sink of -3.3 GgC in the drier year 2017. In 2016, the lakes emitted the largest share of $CH_4$: 66 % considering only diffusive fluxes (Fig. 6c) and 86 % considering total $CH_4$ fluxes (Fig. 6d). Considering the global warming potential of $CH_4$ (IPCC, 2013), $CH_4$ was responsible

for 17% of the total 260 GgCO$_2$eq yr$^{-1}$ emitted in 2016.

### 3.4 Lateral carbon transport

The annual import of carbon to the apex of the Delta amounts to $8490 \pm 240$ GgC yr$^{-1}$ (Fig. 8). This flux consists mostly of inorganic carbon (DIC, 91 %), while DOC and POC comprise only small fractions of 6 and 3 %, respectively. Lateral fluxes are highest in spring, when discharge is highest. About 10% of the Danube's water is channeled into the delta before reaching the Black Sea (Oosterberg et al., 2000), we thus assume that 10% of the annual carbon load of the Danube reaches the delta (i.e. 849 GgC yr$^{-1}$).

The water export from the delta, however, is poorly constrained. The balance between precipitation minus evaporation is negative, poorly quantified and quite variable. We therefore rely on the flux balance of the three branches to estimate carbon export from the delta. The resulting export to the Black Sea via the Danube's main branches amounts to $8650 \pm 150$ GgC yr$^{-1}$ and is less than 2 % higher than the inflow load reaching the apex of delta. The slightly higher load mainly relates to increased DOC levels reaching the main branches from the delta, especially during the spring flood in March and April. The relatively small fraction of water that passes through the delta changes the relative fraction of DOC and POC only marginally to 7 % and 4 %, respectively, while the largest fraction in the water reaching the Black Sea remains DIC (89 %, Fig. 8). DIC import and export is fairly comparable throughout the year, while POC export to the Black Sea strongly exceeded the imports from the catchment in April. DOC exports are highest in the first half of the year (see Fig S5).

## 4 Discussion

### 4.1 The main waterscapes of the Danube delta

As we had hypothesized, carbon dynamics differed significantly across the three different waterscapes. The non-parametric Kruskal-Wallis test followed by the Dunn-Sidak test showed that the median of the three classes are significantly different for concentrations of $CH_4$, $CO_2$, $O_2$ and DIC (see SI). In the case of DOC, only the rivers differ significantly from the other two groups, while in the case of POC, only channels are significantly different. Rivers and lakes, however, may differ significantly in the quality of their POC, as observed by the seasonality of the signal, which shows that high POC in the river actually occurs during high discharge in spring, while high POC in the lakes occurs during algal blooms in late summer. The non-parametric Kruskal-Wallis test does not require normal distribution of the data, but it requires equal variance of the data groups investigated for difference in median (Hedderich and Sachs, 2016). Our observations in the seasonal plots (Fig. 3 & 4) support the results of the test: in most cases, the boxplots do not overlap, indicating that the three groups are significantly different. For example, DOC is significantly higher in the delta lakes and channels due to the strong primary productivity of these systems. $O_2$ is significantly lower in the channels than in the other two categories due to lateral inflow of oxygen-depleted waters from the wetland (Zuijdgeest et al., 2016; Zurbrügg et al., 2012). The large difference between the waterscapes with

respect to $CO_2$ and $CH_4$ fluxes supports our approach to treat the waterscapes independently when upscaling the flux measurements to the total water surface of the delta.

## 4.2 Dominating processes

### 4.2.1 River branches

The main river branches of the Danube are mostly influenced by the hydrology and chemistry of the catchment, as shown by the comparison between the concentrations at the delta apex with concentrations in the three main branches close to the Black Sea. There is comparably little variation between the stations with respect to DIC, DOC and POC. At all sites, $O_2$ is slightly undersaturated most of the times, but we do not see a strong influence of the delta close to the Black Sea.

### 4.2.2 Channels

Carbon dynamics in the channels is strongly affected by the water source. The channels are connecting the river branches to the delta lakes. The direction of this connection depends primarily on hydrologic gradients between the delta and the main branches, which means that flow direction can reverse in individual channels and thus alter their chemical signature due to a change in the main inflow. Seasonally, the channels transport dissolved carbon into the delta and provide nutrients to the reed stands during the high-water season. During times of receding water levels in the main branches, the channels act as the delta's

drainage pipes. The comparison between $CO_2$ fluxes and local $CO_2$ production rates (Fig. 5) shows that the high $CO_2$ fluxes in the channels are often not sustained by in-stream respiration alone, in contrast to what we observed in the river and lakes. While this discrepancy is mainly occurring during high discharge in spring, it is most evident at station 10, where it occurs throughout the year 2016. Station 10 is located in Canalul Vatafu-Imputita, at the border of a core protection zone of the biosphere reserve. During this study, it stood out as a $CO_2$ hot spot, responsible for the highest $CO_2$ concentrations (Fig. 4f).

Additional $CO_2$-rich water inflows from adjacent wetlands could explain the large $CO_2$ fluxes in excess of $CO_2$ production. The water at station 10 was always exceptionally clean, low in oxygen content and had a low pH, supporting the hypothesis of a pronounced input from the reed beds. During times of unusually low water levels, such as in August and September 2017, the lateral influx from the reed seems to cease (Fig. 5). The at first glance contradictory timing of increased lateral inflow during increasing water levels at the other channel stations could be explained by a pressure wave: water flooding the vegetated

area in the west will push out "old" water with a long residence time in the vegetated area at the other edges further east. In general, channel water in the Danube Delta is therefore a mixture of three main sources: Danube river water, lake water and water infiltrating from the wetland. The importance of the individual water source depends on the location of the channel sampling sites and on the water levels, which trigger flooding or draining conditions.

### 4.2.3 Lakes

In the lakes, residence times of 10–30 days allow primary production and local decomposition of organic matter to become important factors driving carbon cycling. We observed abundant macrophytes like *Ceratophyllum demersum* and *Elodea canadensis* growing in spring and early summer, which, depending on lake depth, even reached the lake water surface. A change in abundance of submerged vegetation to vegetation with floating leaves might be linked to changes in the $CO_2$ and $CH_4$ fluxes (Grasset et al., 2016). Around July, algal blooms coincided with a significant reduction in macrophyte abundance.

This pattern seems to be reoccurring due to the eutrophic state of the delta lakes (Tudorancea and Tudorancea, 2006; Coops et al., 2008; Coops et al., 1999). During our observations, both macrophytes and algal blooms caused a drawdown of $CO_2$ and supersaturation in $O_2$ (Fig. 4g & 4h). The algal blooms also partly explain the peak in measured POC from July to November, which extended to most of the delta's channels (Fig. 3f). The degradation of the macrophyte biomass coincided with locally elevated $CH_4$ concentrations from July to October (Fig. 4d).

In constructed wetlands, macrophytes were found to influence the composition of methanogenic communities by affecting dissolved $O_2$ and nitrogen in the rhizosphere, which had a direct impact on the amount of $CH_4$ released to the atmosphere (Zhang et al., 2018). *Potamogeton crispus*, for example, which is also found in the delta lakes and channels, seasonally sustained $CH_4$ fluxes that were up to 3 times higher than $CH_4$ fluxes from *Ceratophyllum demersum* (Zhang et al., 2018). Studies showed, that the plant community composition in the delta lakes shifted since the 1980s due to increasing

eutrophication, which also lead to an increase in *Potamogeton* species recorded in the delta (Sarbu, 2006). It remains unresolved whether this change in vegetation also affected the $CH_4$ release in the Danube Delta.

### 4.3 Uncertainties linked to the upscaling procedure

### 4.3.1 Spatial heterogeneity

    In a hydrologically complex system like the Danube Delta, upscaling $CO_2$ and $CH_4$ is prone to several sources of uncertainties,

most of them linked to the delta's small channels and lakes. First, the channel category showed a large range not only in DOC and POC concentrations, but also in dissolved gases and their fluxes. We attribute this primarily to the varying contribution from the three different water sources, with lateral influx from the reed stands drastically increasing local $CO_2$ concentration and fluxes. One could thus argue that this group is too broad and should be refined. However, in a complex system like the Danube Delta, this is a laborious task, since individual channels are known to reverse flow direction (Irimus, 2006) and

potentially also the amount of lateral inflow depends on the hydrologic conditions in the main branches. The existing 1D-hydrological model "Sobek" (DaNUbs, 2005) could assist in delineating periods of reversed flow, but the detailed model for the exchange with the wetlands would have to be developed.

Second, the surface area of the channels is estimated based on the channel length given in Oosterberg et al. (2000) and an assumed channel width of 19 m, which leads to an estimated surface area that we consider quite conservative. More exact

mapping or better spatial data, which might exist with local authorities but was not at our disposal, could improve this estimate. A larger or smaller surface area attributed to the channel would influence the flux estimates from this category accordingly. Third, we identified station 10 as a $CO_2$ hot spot with concentrations reaching up to 22000 ppm during our study. The hot spot channel had an east to west orientation and was draining a core protection zone. Considering channels with these two criteria indicates potential hot spots could account for up to 2 % of the channel length (see SI) and contribute up to 20 % of the $CO_2$ and $CH_4$ fluxes of the channel category. The overall emissions from the channels (incl. hotspot channels) was decreased by 10 to 30% in this scenario, since considering the high fluxes separately lowered the median value used for the calculation of the channel fluxes. A first step to improve the upscaling would thus be to map the spatial distribution of dissolved gases in the delta. This would give insight on important questions linked to the hot spots: How many hot spots did we miss with our discrete sampling approach? What is their lateral extent? And how steep are the concentration gradients between hot spots and nearby sites?

Fourth, our study neglected small, hardly accessible and remote lakes. A study of various-size lakes in northern Quebec revealed a strong, negative relation between lake $CO_2$ concentration (and fluxes to the atmosphere) and lake area suggesting higher $CO_2$ emission potential of smaller lakes compared to large-area lakes (Marchand et al., 2009). Previous studies of the small area lakes in the Danube Delta characterize them as very clear-water lakes (Coops et al., 1999) with little or no surface water connection to the main branches (Coops et al., 2008), with increased water residence times and $O_2$ concentrations below 5 mg $L^{-1}$ during midday (Oosterberg et al., 2000). This indicated that these lakes, like the hotspot channel in this study, receive the majority of their water from the reed stands (Oosterberg et al., 2000). In contrast to the channels, which are wind sheltered by 2–4 m high reed stands, these small lakes provide a larger surface area and thus a larger wind fetch. Depending on the primary productivity in these lakes, better wind fetch in combination with water contributions from the reed could result in higher fluxes to the atmosphere, at least compared to the larger lakes measured in this study. Based on literature research, we estimate the area of potentially isolated lakes to 99 $km^2$. Attributing these isolated lakes with channel like flux properties would raise the total $CO_2$ and $CH_4$ emissions of the lakes by several fold and turn them from a potential $CO_2$ sink into a $CO_2$ source in 2017 (see SI). The scenario as such represents an extreme case, but it highlights the potentially large contribution from small, so far overlooked lakes in the delta.

**4.3.2 Seasonality**

Seasonal data coverage is often not sufficient to address the seasonality of the fluxes, which might bias the estimates towards either higher or lower emissions. However, not only underrepresentation of certain seasons or events, also the pooling of the data during the upscaling process influences the resulting estimates. In the following, we look at the effects of data pooling for our 2-year data set by comparing different upscaling approaches. In addition to the approach presented in Eq. (4), where we discriminate by year, month and waterscape (case "c"), we also calculated the yearly fluxes in more simple ways by either pooling all data (case "pooled"), discriminating between years only (case "a"), and by discriminating according to year and waterscape (case "b"). In case "c", where we considered individual months, data coverage of $CH_4$ did not allow the calculation

for 2017. In all approaches, we treated the reed stands in the wetlands as a terrestrial part of the system, i.e. excluding them from the analysis.

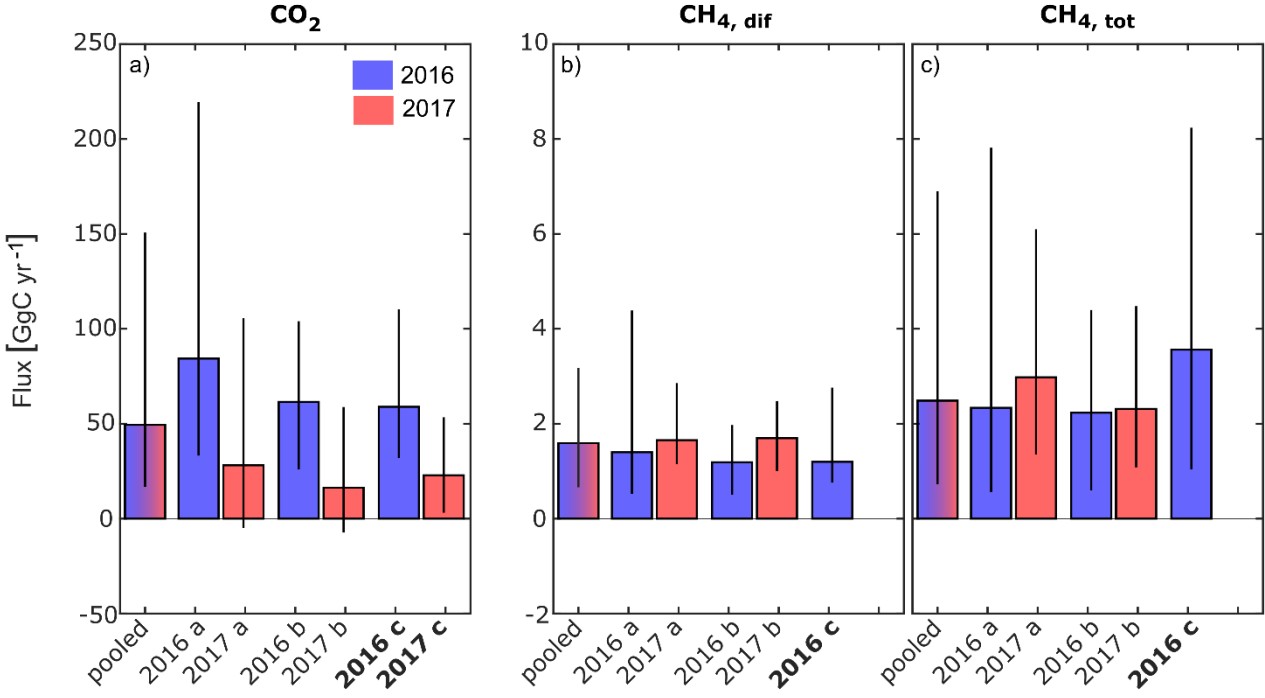

**Figure 7 Comparison of greenhouse gas fluxes from the deltas freshwaters to the atmosphere obtained by the different upscaling approaches "pooled" and cases "a" (discrimination by year), "b" (discrimination by year and waterscape) and "c" (discrimination by year, waterscape and month). Black vertical lines indicate the uncertainty when performing calculations using 25 and 75 percentiles instead of median values. Panel a) $CO_2$ flux, b) diffusive and c) total $CH_4$ flux. All fluxes are in GgC yr$^{-1}$. Bold y-axis labels indicate the calculation approach (case "c") shown in more detail in Figure 6 for the individual contributions from rivers, channels and lakes.**

For the Danube Delta, $CO_2$ flux estimates decreased when considering spatial heterogeneity and seasonality, because the channel data, which showed the most pronounced seasonality and the highest fluxes, are treated independently and assigned to a comparably small area. Independent consideration of data from different years allows exploration of the inter-annual variability, which is quite pronounced for $CO_2$ (Fig. 7a). $CH_4$ emissions tend to be higher in 2017, but the trend is not as clear, especially considering total fluxes (Fig. 7c, case "b"). The lower $CO_2$ flux in 2017 can be explained by the weaker connection of the wetland to the freshwater system of the Danube. We expect that in 2017 most of the water exchange, especially during low discharge conditions, between the river and the inner delta was along the channels as surface water connections, with comparably little water laterally bypassing through the wetland. While the $CO_2$ fluxes from the river were only marginally smaller than in 2016, channels emitted less than 50 % and the lakes even turned into a net $CO_2$ sink in 2017 (Fig. 6a & 6b). The importance of flooded vegetated area on $CO_2$ concentration in rivers was also found in the Congo and Amazon river basins

(Borges et al., 2015; Borges et al., 2019; Amaral et al., 2019), where larger inundated areas correlated with higher $pCO_2$ values. In the case of the lakes, reduced lateral inputs from adjacent wetlands reveal their large $CO_2$ uptake potential. However, this might result in higher $CH_4$ emissions, as calculations according to case "b" indicate. Neglecting seasonality, diffusive $CH_4$ fluxes from the lakes were 0.3 GgC yr$^{-1}$ higher in 2017 (1.0 GgC yr$^{-1}$, data not shown).

Durisch-Kaiser et al. (2008) found Danube lakes to be sources of $CO_2$ and $CH_4$ to the atmosphere in both May and September 2006. Their measured fluxes fall well within the range of our observations. The comparison of data for corresponding months shows, however, that their $CO_2$ concentrations in May are on average twice as high as the ones we measured in 2016, while September concentrations are on average 18 % smaller. The higher fluxes in May could have been due to the aftermath of the severe flood, which reached Romania in the second half of April 2006 and inundated large parts of the delta, thereby promoting lateral exchanges.

## 4.4 Lateral and atmospheric carbon fluxes

The freshwaters of the Danube Delta export in total about 225 GgC yr$^{-1}$ (Fig 8). About 40 % of this carbon is directly released to the atmosphere, while 60 % of the carbon is transported laterally to the Danube and subsequently to the Black Sea. However, the majority of the carbon reaching the Black Sea originates from the catchment (8490 ± 240 GgC yr$^{-1}$). The contribution from the delta is therefore comparably small and the fraction of dissolved and particulate carbon species is only marginally changed by the delta. The anthropogenic alterations of the river main branches, like the straightening and deepening to allow for commercial navigation, might be an explanation for this. Especially the Sulina and the St. George branch were strongly altered in that respect, which has increased the discharge along these branches and decreased the lateral exchange with the delta. Excavation of the channels furthermore increased the surface water connection between different features of the delta.

Considering the area between the three main branches ($A_{Delta}$ = 3510 km$^2$, Table 1) and the catchment area ($A_{catchment}$ = 817000 km$^2$, Table 1), the deltaic carbon yield amounts to 46 gC m$^{-2}$ yr$^{-1}$, while the riverine carbon yield to the Black Sea is 11 gC m$^{-2}$ yr$^{-1}$. So although the Danube Delta contributes only about 2 % to the total carbon load reaching the Black Sea, its role as a carbon source should not be underrated, as the carbon yield (net export/surface area) of the delta is about 4-fold higher than the yield of the overall catchment.

In total, the Danube River and its delta supplied the Black Sea with 8650 ± 150 GgC yr$^{-1}$ in 2016, which fuels carbon emissions in the river plume. Based on concentration measurements in July 1995, Amouroux et al. (2002) estimated the $CH_4$ flux from the Danube River plume close to the St. George branch to 0.47 mmol m$^{-2}$ d$^{-1}$, which compares very well with the $CH_4$ flux we measured in the Danube River branches. As $CH_4$ concentrations in the river plume were 5 to 10 times higher than in the rest of the water column, the authors expect this flux to be fueled by the carbon reaching the Black Sea from the delta. They estimate the total $CH_4$ emissions from river plumes in the Black Sea to be 28–52 GgC yr$^{-1}$, based on the total surface area of the plumes. Since the Danube River is providing more than 50 % of the total discharge and thus is the largest freshwater contributor to the Black Sea (BSC, 2008), the majority of this emission might be released from the Danube River plume. Assuming a share of 50 % of the total river plume emissions would mean that 8–16 % of the carbon laterally transported to

585 the Black Sea might reach the atmosphere in the form of $CH_4$. This corresponds approximately to the share of DOC and POC transported to the Black Sea.

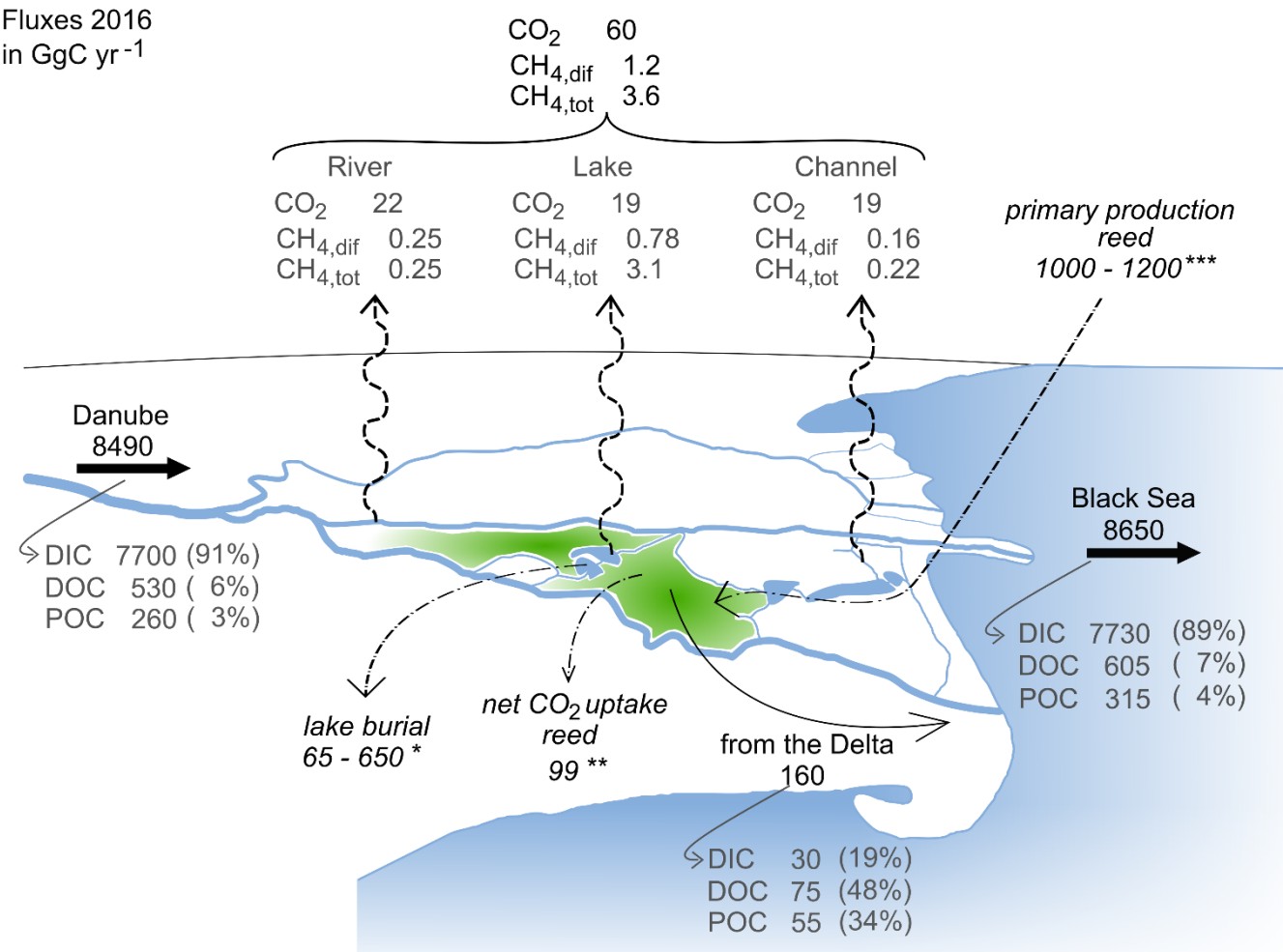

**Figure 8 Overview of carbon flux estimates in GgC yr$^{-1}$. The total area between the main branches is 3510 km$^2$ (see Table 1). Black and grey numbers refer to fluxes estimated during this study based on data from 2016. Italic values refer to estimates based on**
**literature data from different study periods (studies for carbon burial and primary production do not explicitly consider seasonality): *carbon burial in lakes, based on average sedimentation rate measured in 7 lakes in the Danube Delta with an organic carbon content range of 3 – 30 % (Begy et al., 2018), ** net $CO_2$ uptake of *phragmites australis* upscaled to the area covered by scripo-phragmitetum plant community (Zhou et al., 2009), *** upscaled primary productivity of scripo-phragmitetum plant community (Sarbu, 2006). The green area in the plot symbolizes the reed area without indicating all locations of its occurrence.**

The comparison of our lateral DOC and POC fluxes (see Table 3) with available estimates of lateral carbon transport of European rivers to the ocean (Ludwig et al., 1996; Dai et al., 2012), indicates that about 3% and 4% of the POC and DOC could be exported by the Danube River alone. On a global scale, the lateral export of POC compares to the amount exported by the Zambezi River (Teodoru et al., 2015) but is about 20 % lower than the export from the Nile, despite the much higher discharge (Meybeck and Ragu, 1997). Absolute DOC export on the other hand is about twice as high in the Danube compared

to Zambezi and Nile (Teodoru et al., 2015; Badr, 2016). Differences in DOC and POC export are strongly correlated to catchment area or river discharge, while factors such as climate, forest cover, population density or seasonality also affect the respective export fluxes (Alvarez-Cobelas et al., 2012; Hope et al., 1994). Looking at the organic carbon export yields (see Table 4), we observe that this general trend also prevails for the selected rivers, yet the DOC yield of the Danube's catchment surpasses the one of the Mississippi. This might be due to the lower population pressure and lesser agricultural usage of the

Danube Delta, potentially resulting in a better connection of the floodable land to the river. DIC yield, however, is strongly influenced by the lithology of the catchment via silica and carbonate weathering (Gaillardet et al., 1999). The DIC yields of the Mississippi and the Danube catchment, where siliciclastic and carbonate rocks are abundant are also highest, especially in comparison to the Amazon, where a Precambrian basement covers a large part of the heavily weathered catchment. This might explain why the Danube is transporting as much as 1/3 of the Amazon's DIC load, while only having 3% of its discharge

(Moquet et al., 2016; Druffel et al., 2005).

**Table 3 Selected major rivers and their carbon fluxes to ocean and atmosphere**

| River | Export to Ocean [GgC yr$^{-1}$] | | | Water-air flux from the delta [GgC yr$^{-1}$] | | [mmol m$^{-2}$ d$^{-1}$] | |
|---|---|---|---|---|---|---|---|
| | DOC | POC | DIC | $CO_2$ | $CH_4$ | $CO_2$ | $CH_4$ |
| Amazon | 37600 [a] | 6100 [a] | ~24000 [o] – ~30000 [p] | 28500 [e] | 18.7 [f] | 200–1470 [e] | 0.38 [f] |
| Mississippi | 930 [l] – 1900 [c] | 1100 [c] – 3100 [m] | 16000 [i] | | | 55.5 ± 7.6 [i] | |
| Danube | 605 [q] | 315 [q] | 7730 [q] | 60 [q] | 3.6 [q] | 5.8– 93 [q] | 0.42–2.0 [q] |
| Zambezi | 263 [b] | 306 [b] | 3672 [b] | 2731 [b] | 48 [b] | 58.9 [b] | 1.03 [b] |
| Nile | 300 [c, k] | 400 [c] | 12500 [j] | | | | |
| Global | 200000 [n] – 240000 [d] | 240000 [d] – 250000 [n] | 410000 [d] – 450000 [n] | 270000 [g, h] | 709–1800 [h] | 58 [h] | 0.73–1.05 [h] |

[a] Coynel et al. (2005), [b] Teodoru et al. (2015), [c] Meybeck and Ragu (1997), [d] Li et al. (2017), [e] Sawakuchi et al. (2017), [f] flux from Sawakuchi et al. (2014), area for upscaling from Sawakuchi et al. (2017), [g] Laruelle et al. (2010), [h] Borges and Abril (2011), [i] Jiang et al. (2019) total DIC flux estimated using same discharge as [c], [j] Soltan and Awadallah (1995) total flux estimated using same discharge as [c], [k] Badr (2016) total flux estimated, [l] Bianchi et al. (2007), [m] Bianchi et al. (2004), [n] Kirschbaum et al. (2019), [o] Moquet et al. (2016) estimated from $HCO_3^-$ flux, [p] Druffel et al. (2005) total flux estimated using same discharge as [a], [q] this study

**Table 4 annual discharge, catchment area and carbon yields of selected major rivers**

| River | discharge [km³ yr⁻¹] | Catchment area [10⁶ km²] | calculated yields [a] [gC m⁻² yr⁻¹] DOC | POC | DIC |
|---|---|---|---|---|---|
| Amazon | 5444 [b] | 6.4 [c] | 5.9 | 0.95 | 3.8-4.7 |
| Mississippi | 552 [b] | 3.0 [c] | 0.31-0.63 | 0.37-1.0 | 5.3 |
| Danube | 213 [d] | 0.82 [e] | 0.74 | 0.39 | 9.5 |
| Zambezi | 119 [f] | 1.3 [c] | 0.20 | 0.24 | 2.8 |
| Nile | 55.5 [g] | 2.9 [c] | 0.10 | 0.14 | 4.3 |

[a] yield calculated based on catchment area and lateral carbon flux to the Ocean (see Table 3). [b] Dai et al. (2009), [c] Meybeck and Ragu (1997), [d] ICPDR (2018), [e] Tudorancea and Tudorancea (2006), [f] "average literature value" as cited by Teodoru et al. (2015), [g] Badr (2016)

$CO_2$ concentration in large rivers positively correlates with DOC concentration (Borges and Abril, 2011), which can be explained both by simultaneous lateral inputs and by terrestrial organic matter degradation in these net heterotrophic systems. For the selected rivers, the positive correlation also roughly holds for the $CO_2$ fluxes. The $CO_2$ fluxes per unit area from the Danube are much smaller than the ones from the Amazon, but they are closer to those observed in the Mississippi, the Zambezi and the average deduced for estuarine systems (Jiang et al., 2019; Borges and Abril, 2011). Based on this correlation we would expect the $CO_2$ fluxes per unit area for the Nile to be somewhere between the ones from the Amazon and the Zambezi (see Table 3). Sites with high $CO_2$ concentrations are also likely to have a high $CH_4$ content. However, the relation is more complex and not always straightforward (Borges and Abril, 2011). The $CH_4$ fluxes per unit area in the Danube Delta were comparable with those of the Zambezi River but exceeded the fluxes of the large Amazons' inner estuary reported by Sawakuchi et al. (2014).

**4.5 The role of the wetland**

Based on a literature review, Cai (2011) suggested that estuarine $CO_2$ degassing is strongly supported by microbial decomposition of organic matter produced in adjacent coastal wetlands: while $CO_2$ produced in marsh areas and transported to the estuaries was lost to the atmosphere, riverine DIC and DOC content were not greatly altered. Also, several other studies highlight the impact of lateral input of wetlands or floodplain-derived water on river water $O_2$ content (Zurbrügg et al., 2012) and in-stream $CO_2$ levels (D'Amario and Xenopoulos, 2015). Abril and Borges (2019) recently suggested that the active pipe concept of carbon transport in the aquatic continuum indeed needs to be extended to consider floodable and non-floodable land as separate carbon sources. This is in agreement with the present study highlighting how exchange with the wetland can

raise $CO_2$ fluxes well above locally sustained in-stream respiration. In the following, we therefore assess the potential role of the wetland in this complex hydrological system.

The Danube Delta is dominated by the plant association Scirpo-Phragmitetum, which covers nearly 89% of the total marsh area (1600 km$^2$). Its productivity ranges between 1500–1800 g m$^{-2}$ yr$^{-1}$ (Sarbu, 2006). Assuming a carbon content of 0.42 gC gBiomass$^{-1}$ determined by Greenway and Woolley (1999) for *Phragmites australis*, primary production in the reed amounts to 1000–1210 GgC yr$^{-1}$ (Fig. 8), which is about 8 times less than the carbon load transported by the river. A large fraction of the net carbon assimilation by the *phragmites* stands is decomposed and released back to the atmosphere. In a

Danish wetland, more than 50 % of the carbon was respired and released back to the atmosphere, with 48 % being released as $CO_2$ and 4 % as $CH_4$ (Brix et al., 2001). In the Danube Delta, the 50 % accretion rate would correspond to about 500 gC m$^{-2}$ yr$^{-1}$. However, net primary production and carbon accretion change seasonally with environmental factors such as temperature and irradiation. Accordingly, net $CO_2$ assimilation in the Danish study was limited to the warm season from April to September, whereas $CO_2$ and $CH_4$ emission occurred during the whole year but with maxima of 0.2 mol C m$^{-2}$ d$^{-1}$ during

July-August. Qualitatively, we observed the same seasonality in $CO_2$ oversaturation in the channels that drain water from the *Phragmites* stands (Fig. 4f). For a *Phragmites australis* dominated wetland in China, at a latitude comparable to the Danube Delta,  Zhou et al. (2009) estimated the annual net uptake of $CO_2$ to 62 gC m$^{-2}$ yr$^{-1}$. Scaled to the area of the Danube Delta, this would result in 99 GgC yr$^{-1}$ remaining in the delta, which is in the same order of magnitude as the total annual input of organic C from the catchment (79 GgC yr$^{-1}$). Similar to the Danish study, also Zhou et al. (2009) did not account for potential

lateral transport of carbon to adjacent water bodies. Our results show that channels in the Danube Delta are receiving carbon from the wetland, with peaks in $CO_2$ and $CH_4$ concentrations that match the maxima in the gross ecosystem production in China. Comparing the estimated carbon fluxes from the channels with the yearly carbon accumulation estimates of the wetland suggests that up to 20% of the latter could be released to the atmosphere via lateral transport, assuming the carbon fluxes from the channel were exclusively sustained by the wetland. With a lag phase of about 3 months, the Danube Delta reed beds release

peak concentrations of DOC and POC during October to November, when the biomass in the reed stands start degrading (Fig. 3d & 3f).

Assessing the amount of carbon input needed to sustain the observed carbon fluxes in the delta by a simple mass balance approach shows that inputs need to be even higher (Eq. 8). For the mass balance, we consider the net export to the Danube River ($F_{Danube}$ = 160 GgC yr$^{-1}$) and to the atmosphere ($F_{atm}$ = 65 GgC yr$^{-1}$) and assume that sedimentation is predominantly

occurring in the lakes of the delta ($F_{sedi}$). Begy et al. (2018) found averaged sedimentation rates in the lakes of the delta in the range of 0.84 g cm$^{-2}$ yr$^{-1}$. Carbon content in the sediment cores ranged between 3–30 %, translating into a carbon burial rate of 65-650 GgC yr$^{-1}$ across all delta lakes. For the purpose of this simple balance, we neglect anthropogenic effects, e.g. removal of fish biomass or burning of the harvested reed areas during winter and potentially associated carbon inputs.

$$F_{In} \approx -F_{Danube} - F_{atm} - F_{sedi} \qquad (8)$$

$$F_{In} \approx 290 \text{ to } 875 \text{ GgC yr}^{-1}$$

Assuming that freshwaters are a net balanced system and these three fluxes represent all major export fluxes suggests that an input of 290–875 GgC yr$^{-1}$ are required to sustain the export to the Danube, the atmosphere and the sediment. Since long-term carbon burial is most likely an order of magnitude smaller (DeLaune et al., 2018) than the decadal sedimentation rate we expect the required input to be rather at the lower end of the determined range. Nevertheless, it still surpasses the potential contribution

from the wetland as estimated above by a factor of 3. This might either indicate an underestimation of the lateral export from the wetland or significant contributions from other sources, such as the forest areas or anthropogenic inputs to the system from fish farms or wastewater. In addition, emergent macrophytes that border both lakes and channels in the delta could play an important role, since they fix carbon directly from the atmosphere but are decomposed in the water column.

**5 Conclusions**

The waterscapes in the Danube Delta differ significantly with respect to their carbon cycling. While the river is mainly influenced by the carbon signal provided by the upstream catchment, carbon loads and especially greenhouse gas concentrations in the channels are strongly affected by lateral inflow from adjacent wetlands. Local primary production and respiration on the other hand dominate the carbon dynamics in the delta lakes. Considering the spatial extent of the three different waterscapes and the seasonality of their effluxes, we estimate that 65 GgC yr$^{-1}$ (range: 30–120 GgC yr$^{-1}$) were emitted

from the delta to the atmosphere in 2016. Considering the small surface area they cover (7 %), channels in general contributed disproportionately to the total flux (30 %). Small lakes without direct connection to the main river could represent similar hotspots for greenhouse gas evasion as the channels. Overall, nearly 8 % of the total flux to the atmosphere was released as $CH_4$, mostly supplied by the lakes. Covering a full annual cycle and discriminating between the three dominant waterscapes of the delta, we reduce the uncertainty linked to seasonal and spatial variability. However, spatial estimates could be further

improved by investigating the extent of hotspots, gradients between discrete sampling stations, the effect of more isolated lakes and channels of the delta and the inter-annual variability, which especially $CO_2$ seems to show.

We estimate that the Danube Delta receives about 850 GgC yr$^{-1}$ from the upstream catchment. The export surpasses these inputs with the net carbon source from the delta to the Black Sea amounting to about $160 \pm 280$ GgC yr$^{-1}$. However, compared to the overall carbon transfer from the Danube catchment ($8490 \pm 240$ GgC yr$^{-1}$) to the Black Sea, the contribution from the

delta is about 2 % and will not significantly alter the bulk carbon composition of the river water. In terms of carbon yield, the contribution from the delta is about 4-fold higher (45.6 gC m$^{-2}$ yr$^{-1}$) than the riverine carbon yield (10.6 gC m$^{-2}$ yr$^{-1}$).

In order to sustain the observed carbon fluxes from Danube Delta freshwaters to the atmosphere and the Black Sea while assuming a net balanced system, a minimum of 290 GgC yr$^{-1}$ would be required to be provided by the wetland realm or other sources within the Danube Delta.

**Code availability**

The Matlab scripts used for the calculations are available upon request.

**Data availability**

The dataset with the measurements presented in this paper, as well as an accompanying metadata file have not been published elsewhere and are available via the ETH Research Collection (https://doi.org/10.3929/ethz-b-000416925, Maier et al, 2020).

**Sample availability**

Not available

**Video supplement**

Not available

**Supplement Link (Copernicus will include it)**

**Team list**

**Authors Contribution**

BW and CT conceptualized the present study. CT led the monthly monitoring campaigns, supported by MSM. MSM was responsible for lab analysis of samples and subsequent data analysis. MSM prepared the figures and drafted both manuscript and supporting information. All authors engaged in discussing and editing the manuscript.

**Competing interests**

The authors declare that they have no conflict of interest.

**Disclaimer**

**Acknowledgements**

The authors thank Till Breitenmoser, Anna Canning, Christian Dinkel, Tim Kalvelage, Patrick Kathriner and Alexander
Mistretta for support during fieldwork and sample analysis in the lab. We also thank Scott Winton for his comments on the
manuscript. This work was supported by the Swiss State Secretariat for Education, Research and Innovation (SERI) under
contract number 15.0068. The opinions expressed and arguments employed herein do not necessarily reflect the official views
of the Swiss Government. The research leading to these results has received funding from the European Union's Horizon 2020
research and innovation program under the Marie Sklodowska-Curie grant agreement No 643052 (C-CASCADES project).
This product includes data licensed from International Commission for the Protection of the Danube River (ICPDR).

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
