# Peer review of "Spatio-temporal variations of lateral and atmospheric carbon fluxes from the Danube Delta"

_Biogeosciences, 2020_

## Referee Comment (RC1) · Anonymous Referee #1 · 8 Sep 2020

Review bg-2020-197

The MS by Maier et al. presents some extensive and original data of C concentrations and CO2 and CH4 fluxes in the Danube delta. This is a well-designed study. The methods are appropriate, the results are well presented and the interpretations are sound. I recommend the publication of this paper. However, I found few weak points that can easily be improve before publication, based on a more detailed analysis of the data and reading of the literature :

In general, the text is insisting more on spatial variations, rather than temporal variations. Most of the calculated flux numbers are annual averages for the two years

of the study. It would be interesting to interpret more precisely these data in relation with seasonal flooding of the wetland (how do flooded areas change seasonally?), the spring/summer primary production in the wetland, eventually the winter C recycling in the wetland... potentially changing the $CO_2$, $O_2$ and $CH_4$ concentrations and air-water fluxes particularly in the channels.

The results of BOD appear in the discussion but not in the result section. A special paragraph in the result section to describe the discrepancies and concordance between in stream respiration and $CO_2$ outgassing would be useful.

It should be made very clear in the discussion that wetland C metabolic and burial fluxes shown in the last figure are from the literature, not necessarily valid for same study period, and that they do not consider seasonal variations.

Line by line comments:

Abstract : I miss some information about seasonal variations please provide standard deviations on flux numbers L21 & 22 L25, explicit what form of C is exported from the delta: is it OC or DIC ?

L43 : the fact Âń carbon inputs from terrestrial ecosystems degas as $CO_2$ and $CH_4$ along the way to the ocean Âż is known for a long time, and not only from Âń recent estimates Âż In the introduction, it is important to cite pioneer papers and not refer all the time to very recent work that only confirmed the previous study, and do not provide any new information about the mentioned statement.

L59 : The statement "riparian wetlands in the Amazon basin have been identified as significant sources for the outgassing of terrestrial carbon in the form of $CO_2$" Cite also Abril et al. 2014 here.

L62: "While wetlands are estimated to contribute 1.1 PgC yr-1 (Aufdenkampe et al., 2011) to the global carbon emissions, Amazonian wetland emission alone could contribute another 0.2 PgC yr-1 (Abril et al., 2014). Specifically, riparian systems in the lowlands could provide significant lateral carbon inputs (Sawakuchi et al., 2017)." These references to the literature are partially inappropriate. There is a confusion here between $CO_2$ outgassing from waters and $CO_2$ emissions from wetland ecosystem. Abril et al. proposed that central amazon wetland + river channel could be at equilibrium (zero flux), the flooded forest and marophytes being a sink and the open waters a source. Also, no need to be so precise on Amazon numbers in an introduction of a MS on the Danube delta.

L68, mention that flooding has been recently described an important transport mechanism of terrestrial C to aquatic system, additional to drainage and surface runoff.

L162: "As tests showed that there was no significant difference between the lab- and field-based methods, we pooled the data in our analysis." I suggest you provide the result of these tests as a figure as supplementary material

L184: "In the high-resolution LGR time series, the influence of gas bubbles could easily be identified." The method is described graphically in Grasset et al. Freshwater Biology. doi:10.1111/fwb.12780.

L242 In section "2.6 Import by Danube River and Export to Black Sea", please provide here or maybe in the results section, more precise verbal information (equation is ok) on how you calculate C lateral fluxes before and after the wetland in the delta and how you deal with the problem that these two fluxes might be too close to each other to allow a precise calculation of the net lateral export from the wetland in the delta as a small differences between two large numbers that contain some uncertainty. What are the representativity of stations and data, with respect to observed spatial and temporal differences in the C forms and discharge data between sampling points.

L290: figure 3 and next ones would be easier to read if simultaneous discharge could be shown

L350 and k values. Lake are more exposed to wind indeed, however rivers and channels are exposed to current which may also contribute to k600

L380 you are repeating what has been said in Mat and met about CH4

L394. The calculation of lateral flux is indeed poorly constrained and it would be interesting to see the data that support the statements "POC import from the catchment exceeds the export to the Black Sea in February and March, while DOC import exceeds export only during August (data not shown)."

L400-410, please simplify the story about statistics.

442: what about summer stratification in lakes?

L445: the study of CH4 emissions from various plant types by Grasset et al 2016 might be helpful here

L584 drylands are not defined as the contrary of wetlands. The difference here is between floodable land and non-floodable land

L594 "Assuming a carbon content of 0.42 gC gBiomass-1 determined by Greenway and Woolley (1999) for Phragmites australis, between 1000 and 1210 GgC yr-1 are bound in the form of macrophyte biomass in the reed area of the Danube Delta". Not clear the meaning of "bound" here. A great portion of the macrophyte biomass is supposed to be recycled. "The wetlands thus hold 12 to 17 times the total input of organic C to the delta from the catchment" do you mean the total ANNUAL input?

L595 "Nevertheless, wetlands are considered to be net C sinks and Zhou et al. (2009) estimate the sink capacity of a Phragmites australis dominated wetland to -62 gC m-2 yr-1 considering CO2 and CH4 release from the wetland itself." You must be more precise here in the vocabulary used. CO2 sink might be different from C sink. Please specify that C sink is OC burial in sediments and not the atmospheric CO2 sink. Same in the following sentences.

L600 "Assuming that carbon emitted from the channels originated only from the wet-

land source, this would suggest that up to 20 % of the potential wetland sink might be exported laterally, eventually finding its way to the atmosphere." Please explain more clearly

---

## Referee Comment (RC2) · Anonymous Referee #2 · 24 Sep 2020

This paper reports results of a 2-year field study of carbon dioxide and methane fluxes from freshwater systems in the Danube Delta. The study focusses on 19 sites and provides insight in seasonally resolved fluxes and lateral carbon transport. This must have been an enormous effort. The authors find that lakes are the largest emitters of methane. Channels show a wide range of emissions and may be hotspots both for carbon dioxide and methane.

The paper is well-written and the results are well-presented. I do find the comparison to fluxes in other rivers rather descriptive. This is where a slightly more process-based comparison could increase the impact of the paper. I have only a few minor other

comments, mostly editorial, see below.

Line 14. What is meant by "reference" systems in this sentence. Could you rephrase?

Line 94. Suggested change: "Station 16 was removed from the study because of limited access".

Line 170. Suggested change: "stored in the dark"

Line 172-173: Suggested change: "production rates"

Line 174. Suggested change: "are underestimating respiration rate" => "underestimate respiration rates"

Line 177. Change to "rates"

Line 235. Specify that you are referring to the width of the channels here. Is there no estimate of the number of these old meanders? I realize you discuss the uncertainty in the channel width later in the manuscript, but this ten-fold difference in width is still rather substantial.

Line 350 "at 0.69" instead of 'with 0.69"

Line 362: suggested change: "cause the" => "contributed to the"

Line 363: suggested change: "seems to be" => "appears to be"

Line 371: this sentence needs rephrasing: "A look into the contribution from the different waterscapes shows that the river branches the main source of CO2 to the atmosphere were in both years"

Line 371: use past tense: "switched"

Line 388: It's not clear what "it" refers to in "It mainly relates to"

Line 397: change to"in the case of"

Line 415. Change to "do not see"

Line 442. Longer when compared to what? Please specify

Line 445. "Seems to be reoccurring" raises the question what evidence there is for that. You might consider rephrasing to "This pattern may be due to the eutrophic state. . .." unless you can be more specific.

Line 514. "data. . .are treated"

Line 553: "and thus is the"

Line 565. The Black Sea has only a limited connection to the open sea. Is it meaningful to include it in the estimate of the DOC and POC flux to the ocean?

Line 565-570. In this section, you are comparing your DIC, POC and DOC fluxes to those of other rivers, but you are not providing any explanations for the observed differences. Adding that would make the comparison more useful.

Lines 572-575: This is again very descriptive. Are there any possible explanations for the observed differences between rivers?

Line 585. Change to "we therefore assess the potential role". Your intentions are not relevant.

Line 599. What does "decal" refer to?

Table 3: typo in the heading of the right column "water-air flux from delta"

Line 632: change to "these"

---

## Author Comment (AC1) · 8 Nov 2020

Dear Anonymous Referee #2,

Thank you very much for your positive feedback, we highly appreciate the time and effort you invested to review our MS. In the following, you find our replies (black, indented) to the *individual comments* (blue, italic).

*This paper reports results of a 2-year field study of carbon dioxide and methane fluxes from freshwater systems in the Danube Delta. The study focusses on 19 sites and provides insight in seasonally resolved fluxes and lateral carbon transport. This must have been an enormous effort. The authors find that lakes are the largest emitters of methane. Channels show a wide range of emissions and may be hotspots both for carbon dioxide and methane.*
*The paper is well-written and the results are well-presented. I do find the comparison to fluxes in other rivers rather descriptive. This is where a slightly more process-based comparison could increase the impact of the paper.*

Thank you for pointing this out. We did discuss responsible processes in the manuscript, but we outline below for the discussion part (lines 565 ff) how to expand the processes-based comparison with other river systems.

*I have only a few minor other comments, mostly editorial, see below.*
*Line 14. What is meant by "reference" systems in this sentence. Could you rephrase?*

By «reference» system, we mean that the Danube River reaches can be used as a point of comparison: the water in the Danube Delta originates mainly from the Danube River, precipitation is only a minor water source. The river water can therefore act as a reference to establish concentration changes in the delta with respect to the water chemistry provided by the catchment. We will define this more clearly:

We plan to edit this as follows:

"In this paper, we use the river reaches that cross the delta as references systems. The water chemistry in the Danube River is defined by upstream processes in the catchment. The effects of biogeochemical processes of primary production and respiration in the lakes and reed stands imprint different chemical signatures on the water in lakes and channels. "

*Line 94. Suggested change: "Station 16 was removed from the study because of limited access".*

Accepted. We will slightly expand the statement: "Station 16 was removed from the study because of limited access during lower water level (clogged access channel)."

*Line 170. Suggested change: "stored in the dark"*

Accepted. The sentence now reads: "The other three bottles were stored in the dark at approximately in-situ temperatures and $O_2$ concentration was measured after 24 hours."

*Line 172-173: Suggested change: "production rates"*

Since the bottles were stored in the dark, we measured a decline in $O_2$, so it would rather be a "$O_2$ consumption rate". In our opinion using the term "production rate" here might be

misleading, since the same experimental set-up storing the bottles in light instead of dark conditions would indeed measure an $O_2$/primary production rate.

The sentence now reads: "The $O_2$ consumption rate was derived from the time and concentration difference, assuming a linear decrease over time."

*Line 174. Suggested change: "are underestimating respiration rate" => "underestimate respiration rates"*

Accepted. The sentence now reads: "Ward et al. (2018)argue that respiration rate measurements in BOD bottles underestimate respiration rate because microbial processes are limited by both the bottle size and the lack of turbulence and suggest a correction factor of 2.7 to correct BOD derived respiration rates for size effects only or a factor of 3.7 for size and low turbulence effects."

*Line 177. Change to "rates"*

Accepted. The sentence now reads: "Applying these correction factors did not change the main point of our comparison between fluxes and $CO_2$ production rates."

*Line 235. Specify that you are referring to the width of the channels here. Is there no estimate of the number of these old meanders? I realize you discuss the uncertainty in the channel width later in the manuscript, but this ten-fold difference in width is still rather substantial.*

There are two big old meanders along the Sulina branch, where stations 11 and 17 are located. They are now bypassed by the rectified, shorter branch of the Danube. Their area is about 4.5 $km^2$. We also calculated the area of the channels based on the length of all channels recorded in a publicly available shape file from mapcruzin.com (2016) and the area of these two meanders. With this approach, using the same average width, we arrive at a channel area of 31 $km^2$ (compared to 33 $km^2$ with the approach used in the paper). The main uncertainty lies in the width and number of the very small channels, that are very difficult to determine from areal photos, while the larger meanders can be measured comparably easy.

Changes for clarification: "Especially the old, cut-off meanders of the Danube River (Dunarea Veche), which we also consider as belonging to the channel category, do have a much larger width ranging in the order of 100–200 m."

Additional comment: we noticed a typo in Table 1 concerning the calculation of the channel area: Instead of 10 m width, we actually used 19 m width for the calculation and will correct this in the revised version: 1753 km * 19 m = 33 $km^2$

*Line 350 "at 0.69" instead of 'with 0.69"*

Accepted. The sentence now reads: "Median $k_{600}$ was lowest in in the river branches and in the channels at 0.69 m $d^{-1}$ and 0.74 m $d^{-1}$, respectively (see Table S1)."

*Line 362: suggested change: "cause the" => "contributed to the"*

Accepted. The sentence now reads: "It is likely that the different hydrological conditions triggered different amounts of lateral inflow from the reed-covered wetlands and cause the large variability in CO2 fluxes."

*Line 363: suggested change: "seems to be" => "appears to be"*

Accepted. The sentence now reads: "For $CH_4$, this effect appears to be much smaller."

*Line 371: this sentence needs rephrasing: "A look into the contribution from the different waterscapes shows that the river branches the main source of $CO_2$ to the atmosphere were in both years"*
*Line 371: use past tense: "switched"*

Corrected sentence: "Considering the contributions from the different waterscapes shows that the river branches were the main source of $CO_2$ to the atmosphere in both years".

Rephrased sentence: "The slightly higher load mainly relates to increased DOC levels reaching the main branches from the delta, especially during the spring flood."

Accepted. The sentence now reads: "In the case of DOC, only the rivers differ significantly from the other two groups, while in the case of POC, only channels are significantly different."

Accepted. The sentence now reads: "At all sites, $O_2$ is slightly undersaturated most of the times, but we do not see a strong influence of the delta close to the Black Sea."

Results from the Sobek model simulation showed that the residence times of the lakes we studied mostly surpassed the travel time of the water to the individual lake (Oosterberg et al., 2000), so the combined time spend in channels and rivers on the way to the lake. This is where the "longer" came from in this sentence. However, we do acknowledge that the water might be spending longer time in stagnant channels (albeit in connection with the wetland) and therefore propose to remove the "longer".

Changed sentence: "In the lakes, residence times of 10–30 days allow primary production and local decomposition of organic matter to become important factors driving carbon cycling."

*Line 445.* "*Seems to be reoccurring*" *raises the question what evidence there is for that. You might consider rephrasing to* "*This pattern may be due to the eutrophic state. . . .*" *unless you can be more specific.*

Tudorancea and Tudorancea (2006) report reoccurring algal blooms in the period from 1977 to 1999 without mentioning the month of occurrence. Two studies (Coops et al., 2008; Coops et al., 1999) based on data from 1996-1998 indicate the timing of the algal blooms was also around July and occurred simultaneously with a decrease in macrophyte abundance.

E.g. Coops et al. (2008): "Comparison between early and late summer vegetation showed a distinct seasonality of the vegetation in the large lakes: these lakes were almost entirely covered by Potamogeton spp. vegetation in June, and devoid of macrophytes in late summer. Concomitantly a strong decrease in transparency had occurred with the development of algal blooms."

The observations of these studies date several years before our study and nutrient levels in the Danube River reduced since then, therefore the phrasing "seems to be reoccurring" - still.

The corrected sentence now reads: "For the Danube Delta, $CO_2$ flux estimates decreased when considering spatial heterogeneity and seasonality, because the channel data, which showed the most pronounced seasonality and the higher fluxes, are treated independently and assigned to a comparably small area."

Corrected sentence: "Since the Danube River is providing more than 50 % of the total discharge and thus is the largest freshwater contributor to the Black Sea …"

*Line 565. The Black Sea has only a limited connection to the open sea. Is it meaningful to include it in the estimate of the DOC and POC flux to the ocean?*

Although the Black Sea is a marginal sea, it is included in studies concerning global riverine carbon or nutrient export to the ocean (e.g. M. Dai et al., 2012; Li et al., 2017; Ludwig et al., 1996). We therefore considered it reasonable to compare our estimates of the organic carbon export flux of the Danube River to the estimated carbon fluxes of European rivers, especially since the Danube is Europe's second largest river.

*Line 565-570. In this section, you are comparing your DIC, POC and DOC fluxes to those of other rivers, but you are not providing any explanations for the observed differences. Adding that would make the comparison more useful.*

For the comparison of the different rivers our intention is to look at the carbon yield of the respective river catchments, which we summarized in the following. We will add this information to Table 3 and expand the discussion as presented below.

| River | Q_km3/yr | Source | Area [10^6 km2] | source | calculated yields [gC/m2/yr] DOC | POC | DIC |
|---|---|---|---|---|---|---|---|
| Amazon | 5444 | A. Dai et al. (2009) | 6.4 | Meybeck and Ragu (1997) | 5.9 | 0.95 | 3.8-4.7 |
| Mississippi | 552 | A. Dai et al. (2009) | 3.0 | Meybeck and Ragu (1997) | 0.31-0.63 | 0.37-1.0 | 5.3 |
| Danube | 213 | ICPDR (2018) | 0.82 | Tudorancea and Tudorancea (2006) | 0.74 | 0.39 | 9.5 |
| Zambezi | 119 | "av lit value" as cited by Teodoru et al. (2015) | 1.3 | Meybeck and Ragu (1997) | 0.20 | 0.24 | 2.8 |
| Nile | 55.5 | Badr (2016) | 2.9 | Meybeck and Ragu (1997) | 0.10 | 0.14 | 4.3 |

"The comparison of our lateral DOC and POC fluxes with available estimates of lateral carbon transport of European rivers to the ocean (M. Dai et al., 2012; Ludwig et al., 1996), indicates that about 3% and 4% of the POC and DOC could be exported by the Danube River alone. On a global scale, the lateral export of POC compares to the amount exported by the Zambezi River (Teodoru et al., 2015) but is about 20 % lower than the export from the Nile, despite the much higher discharge (Meybeck & Ragu, 1997). Absolute DOC export on the other hand is about twice as high in the Danube compared to Zambezi and Nile (Badr, 2016; Teodoru et al., 2015). Differences in DOC and POC export are strongly correlated to catchment area or river discharge, while depending on climate, factors such as forest cover, population density or seasonality also affect the respective export fluxes (Alvarez-Cobelas et al., 2012; Hope et al., 1994). Looking at the organic carbon export yields (see Table 3), we observe that this general trend also prevails for the selected rivers, yet the DOC yield of the Danube's catchment surpasses the one of the Mississippi. This might be due to the lower population pressure and lesser agricultural usage of the Danube Delta, potentially resulting in a better connection of the floodable land to the river.

DIC yield, however, is strongly influenced by the lithology of the catchment via silica and carbonate weathering (Gaillardet et al., 1999). The DIC yields of the Mississippi and the Danube catchment, where siliciclastic and carbonate rocks are abundant are also highest, especially in comparison to the Amazon, where a Precambrian basement covers a large part of the heavily weathered catchment. This might explain why the Danube is transporting as much as 1/3 of the

Amazon's DIC load, while only having 3% of its discharge (Druffel et al., 2005; Moquet et al., 2016)."

*Lines 572-575: This is again very descriptive. Are there any possible explanations for the observed differences between rivers?*

CO$_2$ concentrations in large rivers positively correlates with DOC concentration (Borges & Abril, 2011), which can be explained both by simultaneous lateral inputs and by terrestrial organic matter degradation in these net heterotrophic systems. For the selected rivers, the positive correlation also roughly holds for the CO$_2$ fluxes. The CO$_2$ fluxes per unit area from the Danube are much smaller than the ones from the Amazon, but they are closer to those observed in the Mississippi, the Zambezi and the average deduced for estuarine systems (Borges & Abril, 2011; Jiang et al., 2019). Based on this correlation we would expect the CO$_2$ fluxes per unit area for the Nile to be somewhere between the ones from the Amazon and the Zambezi.

Sites with high CO$_2$ concentrations are also likely to have large CH$_4$ content. However, the relation is more complex and not always straightforward (Borges & Abril, 2011). The CH$_4$ fluxes per unit area in the Danube Delta were comparable with those of the Zambezi River but exceeded the fluxes of the large Amazons' inner estuary reported by Sawakuchi et al. (2014).

*Line 585. Change to "we therefore assess the potential role". Your intentions are not relevant.*

Corrected sentence: "In the following, we therefore assess the potential role of the wetland in this complex hydrological system."

*Line 599. What does "decal" refer to?*

"decal" is a typo in this context and was supposed to be "decadal" and refers to the estimate carbon storage/sedimentation rate. The corrected sentence reads: "In the Mississippi Delta, DeLaune et al. (2018) found long-term storage of wetlands up to one order of magnitude lower than expected from the decadal sedimentation rate."

*Table 3: typo in the heading of the right column "water-air flux from delta"*

Thanks!

*Line 632: change to "these"*

Corrected sentence: "The export surpasses these inputs with the net carbon source from the delta to the Black Sea amounting to about 160 GgC yr$^{-1}$."

**References**

Alvarez-Cobelas, M., Angeler, D. G., Sánchez-Carrillo, S., & Almendros, G. (2012). A worldwide view of organic carbon export from catchments. *Biogeochemistry, 107*(1), 275-293. doi:10.1007/s10533-010-9553-z

Badr, E.-S. A. (2016). Spatio-temporal variability of dissolved organic nitrogen (DON), carbon (DOC), and nutrients in the Nile River, Egypt. *Environmental Monitoring and Assessment, 188*(10), 580. doi:10.1007/s10661-016-5588-5

Borges, A. V., & Abril, G. (2011). Carbon Dioxide and Methane Dynamics in Estuaries. In E. Wolanski & D. McLusky (Eds.), *Treatise on Estuarine and Coastal Science* (pp. 119-161). Waltham: Academic Press.

Coops, H., Buijse, L. L., Buijse, A. D. T., Constantinescu, A., Covaliov, S., Hanganu, J., . . . Oosterberg, W. (2008). Trophic gradients in a large-river Delta: ecological structure determined by connectivity gradients in the Danube Delta (Romania). *River Research and Applications, 24*(5), 698-709.

Coops, H., Hanganu, J., Tudor, M., & Oosterberg, W. (1999). Classification of Danube Delta lakes based on aquatic vegetation and turbidity *Biology, Ecology and Management of Aquatic Plants* (pp. 187-191): Springer.

Dai, A., Qian, T., Trenberth, K. E., & Milliman, J. D. (2009). Changes in Continental Freshwater Discharge from 1948 to 2004. *Journal of Climate, 22*(10), 2773-2792. doi:10.1175/2008jcli2592.1

Dai, M., Yin, Z., Meng, F., Liu, Q., & Cai, W.-J. (2012). Spatial distribution of riverine DOC inputs to the ocean: an updated global synthesis. *Current Opinion in Environmental Sustainability, 4*(2), 170-178. doi:https://doi.org/10.1016/j.cosust.2012.03.003

DeLaune, R. D., White, J. R., Elsey-Quirk, T., Roberts, H. H., & Wang, D. Q. (2018). Differences in long-term vs short-term carbon and nitrogen sequestration in a coastal river delta wetland: Implications for global budgets. *Organic Geochemistry, 123*, 67-73. doi:https://doi.org/10.1016/j.orggeochem.2018.06.007

Druffel, E. R. M., Bauer, J. E., & Griffin, S. (2005). Input of particulate organic and dissolved inorganic carbon from the Amazon to the Atlantic Ocean. *Geochemistry, Geophysics, Geosystems, 6*(3). doi:10.1029/2004gc000842

Gaillardet, J., Dupré, B., Louvat, P., & Allègre, C. J. (1999). Global silicate weathering and CO2 consumption rates deduced from the chemistry of large rivers. *Chemical Geology, 159*(1), 3-30. doi:https://doi.org/10.1016/S0009-2541(99)00031-5

Hope, D., Billett, M. F., & Cresser, M. S. (1994). A review of the export of carbon in river water: Fluxes and processes. *Environmental Pollution, 84*(3), 301-324. doi:https://doi.org/10.1016/0269-7491(94)90142-2

ICPDR. (2018). Danube River Basin Water Quality Database. Retrieved 02.02.2018 http://www.icpdr.org/wq-db/

Jiang, Z.-P., Wei-Jun, C., Lehrter, J., Chen, B., Ouyang, Z., Le, C., . . . Zhang, J. (2019). Spring net community production and its coupling with the CO 2 dynamics in the surface water of the northern Gulf of Mexico. *Biogeosciences, 16*(18), 3507-3525.

Li, M., Peng, C., Wang, M., Xue, W., Zhang, K., Wang, K., . . . Zhu, Q. (2017). The carbon flux of global rivers: A re-evaluation of amount and spatial patterns. *Ecological Indicators, 80*, 40-51. doi:https://doi.org/10.1016/j.ecolind.2017.04.049

Ludwig, W., Probst, J.-L., & Kempe, S. (1996). Predicting the oceanic input of organic carbon by continental erosion. *Global Biogeochemical Cycles, 10*(1), 23-41. doi:10.1029/95gb02925

mapcruzin.com. (2016, 13.01.2016). Retrieved from https://mapcruzin.com/free-romania-arcgis-maps-shapefiles.htm, based on www.openstreetmap.org/

Meybeck, M., & Ragu, A. (1997). *River discharges to the oceans: an assessment of suspended solids, major ions and nutrients* (Vol. 245): UNEP.

Moquet, J. S., Guyot, J. L., Crave, A., Viers, J., Filizola, N., Martinez, J. M., . . . Pombosa, R. (2016). Amazon River dissolved load: temporal dynamics and annual budget from the Andes to the ocean. *Environ Sci Pollut Res Int, 23*(12), 11405-11429. doi:10.1007/s11356-015-5503-6

Oosterberg, W., Staras, M., Bogdan, L., Buijse, A. D., Constantinescu, A., Coops, H., . . . Navodaru, I. (2000). Ecological gradients in the Danube Delta lakes: present state and man-induced changes.

Sawakuchi, H. O., Bastviken, D., Sawakuchi, A. O., Krusche, A. V., Ballester, M. V., & Richey, J. E. (2014). Methane emissions from Amazonian Rivers and their contribution to the global methane budget. *Global Change Biology, 20*(9), 2829-2840.

Teodoru, C. R., Nyoni, F. C., Borges, A. V., Darchambeau, F., Nyambe, I., & Bouillon, S. (2015). Dynamics of greenhouse gases (CO 2, CH 4, N 2 O) along the Zambezi River and major tributaries, and their importance in the riverine carbon budget. *Biogeosciences, 12*(8), 2431-2453.

Tudorancea, C., & Tudorancea, M. M. (2006). *Danube Delta: genesis and biodiversity*. Leiden: Backhuys Publishers.

Ward, N. D., Sawakuchi, H. O., Neu, V., Less, D. F. S., Valerio, A. M., Cunha, A. C., . . . Keil, R. G. (2018). Velocity-amplified microbial respiration rates in the lower Amazon River. *Limnology and Oceanography Letters, 3*(3), 265-274. doi:doi:10.1002/lol2.10062

---

## Author Comment (AC2) · 8 Nov 2020

Dear Anonymous Referee #1,

Thank you very much for your constructive comments regarding our MS, we highly appreciate the time and effort you invested. In the following, you find our replies (black, indented) to your *individual comments* (blue, italic).

*The MS by Maier et al. presents some extensive and original data of C concentrations and CO2 and CH4 fluxes in the Danube delta. This is a well-designed study. The methods are appropriate, the results are well presented and the interpretations are sound. I recommend the publication of this paper. However, I found few weak points that can easily be improve before publication, based on a more detailed analysis of the data and reading of the literature: In general, the text is insisting more on spatial variations, rather than temporal variations. Most of the calculated flux numbers are annual averages for the two years of the study. It would be interesting to interpret more precisely these data in relation with seasonal flooding of the wetland (how do flooded areas change seasonally?), the spring/summer primary production in the wetland, eventually the winter C recycling in the wetland: potentially changing the $CO_2$, $O_2$ and $CH_4$ concentrations and air-water fluxes particularly in the channels. The results of BOD appear in the discussion but not in the result section. A special paragraph in the result section to describe the discrepancies and concordance between in stream respiration and $CO_2$ outgassing would be useful. It should be made very clear in the discussion that wetland C metabolic and burial fluxes shown in the last figure are from the literature, not necessarily valid for same study period, and that they do not consider seasonal variations.*

It was our intention in this paper to focus on the functional differences between lakes and channels. A detailed analysis of the seasonal variability in relation to the flooded area would require a detailed remote-sensing study and this was beyond the scope of this paper. Also, we did not have the capacity to perform direct primary productivity measurements of the reed stands, which would have required a rather complex study design with aquatic plus terrestrial observations. We welcome, however, the other suggestions:

- In the discussion section, we will now include literature data that report the seasonal cycle of reed growth and decomposition. See detailed text below (L600).

- In the revised version, we will also add a short paragraph to the results section documenting the relations between stream respiration and outgassing. Fig. 6 will also be shifted to this section.

**"3.2.3 $CO_2$ production vs. $CO_2$ flux**

We find respiration rates ranging between 0.8–390 mM m$^{-2}$ d$^{-1}$ for rivers, while in the channels and lakes they ranged from 2.3–560 mM m$^{-2}$ d$^{-1}$ and 1.0–350 mM m$^{-2}$ d$^{-1}$, respectively (Fig. 6 and Fig. S5-S7). Median respiration rate is highest in rivers (54 mM m$^{-2}$ d$^{-1}$), followed by lakes (48 mM m$^{-2}$ d$^{-1}$) and channels (45 mM m$^{-2}$ d$^{-1}$). Many stations showed a pronounced seasonality with highest respiration rates occurring mostly

between July to October. Respiration rates, i.e. $CO_2$ production rates generally exceed $CO_2$ fluxes in river and lake stations throughout the year (Fig. 6), which implies that local instream $CO_2$ production sustained the observed fluxes. At the channel stations we frequently observed fluxes exceeding the local production, even if we account for potential underestimation of the $CO_2$ production, which implies the presence of other $CO_2$ sources. This was most striking at station 10, the $CO_2$ hotspot, where $CO_2$ outgassing exceeded local respiration on average by a factor of 40. At the other channel stations (also see Fig S6), there seems to be a seasonally occurring pattern: $CO_2$ fluxes exceed local production in the first half of the year, while for the remainder of the year they fall below. While this pattern is very distinct in 2016, it is less pronounced in the drier year 2017, which suggests that the additional $CO_2$ source is linked to hydrology."

- Finally, we will point out more clearly in Fig. 8, that the rates of burial and wetland metabolism were taken from the literature with different inherent timescales. For this purpose, we will expand the figure caption accordingly:

"Italic values refer to estimates based on literature data from different study periods (carbon burial and net primary production do not explicitly consider seasonality): *carbon burial in lakes, based on average sedimentation rate measured in 7 lakes in the Danube Delta with an organic carbon content range of 3 – 30 % (Begy et al., 2018), ** sink capacity of *phragmites australis* upscaled to the area covered by scripo-phragmitetum plant community (Zhou et al., 2009), *** upscaled net primary productivity of scripo-phragmitetum plant community (Sarbu, 2006)."

**Line by line comments**

*Abstract: I miss some information about seasonal variations please provide standard deviations on flux numbers L21 & 22 L25, explicit what form of C is exported from the delta: is it OC or DIC?*

L21 & L22: As our data is not normally distributed, we are reporting median concentrations for the $CO_2$ and $CH_4$ fluxes to the atmosphere from the different compartments. Providing standard deviations along with the median values would be inconsistent. In the revised version we will provide standard deviations for the overall annual fluxes of GHG including the ranges we obtain from calculations with the 25 and 75 percentile: "65 Gg C yr$^{-1}$ (30–120 Gg C yr$^{-1}$, range calculated using 25–75 percentile of observed fluxes)"

L25: The number refers to the total export of carbon, i.e. the sum of OC and DIC. We add the standard deviations calculated using gaussian error propagation and define the kind of carbon exported: "In terms of lateral export, we estimate the net total export (DIC+DOC+POC) from the Danube Delta to the Black Sea to about 160 ± 280 GgC yr$^{-1}$, which only marginally increases the carbon load from the upstream river catchment (8490 ± 240 GgC yr$^{-1}$) by about 2 %. "

*L43: the fact "carbon inputs from terrestrial ecosystems degas as $CO_2$ and $CH_4$ along the way to the ocean" is known for a long time, and not only from "recent estimates". In the introduction, it is important to cite pioneer papers and not refer all the time to very recent work that only confirmed the previous study, and do not provide any new information about the mentioned statement.*

Thanks for the suggestion. While there are different valid approaches to cite recent or more classical literature, we will add a classical reference that already documents the $CO_2$ supersaturation in freshwater ecosystems: Stumm and Morgan (1981) and Cole et al. (2007)

*L59: The statement "riparian wetlands in the Amazon basin have been identified as significant sources for the outgassing of terrestrial carbon in the form of CO2" Cite also Abril et al. 2014 here.*

We will add this reference.

*L62: "While wetlands are estimated to contribute 1.1 PgC yr-1 (Aufdenkampe et al., 2011) to the global carbon emissions, Amazonian wetland emission alone could contribute another 0.2 PgC yr-1 (Abril et al., 2014). Specifically, riparian systems in the lowlands could provide significant lateral carbon inputs (Sawakuchi et al., 2017)." These references to the literature are partially inappropriate. There is a confusion here between $CO_2$ outgassing from waters and $CO_2$ emissions from wetland ecosystem. Abril et al. proposed that central amazon wetland + river channel could be at equilibrium (zero flux), the flooded forest and marophytes being a sink and the open waters a source. Also, no need to be so precise on Amazon numbers in an introduction of a MS on the Danube delta.*

Thank you for these critical remarks. In the revised version, we will restrict the discussion to estimates of the global contribution of wetlands to aquatic emissions and add a note of caution regarding the confounding factors:

"Global wetlands were estimated to contribute 1.1 PgC yr$^{-1}$ (Aufdenkampe et al., 2011) to the carbon emissions in the land-ocean aquatic continuum. The uncertainty of these estimates is large, due to the difficulty to delineate global wetland areas (Tootchi et al., 2019) and the complex interaction between potential emissions and carbon uptake by vegetation and soils (Hastie et al., 2019)"

*L68, mention that flooding has been recently described an important transport mechanism of terrestrial C to aquatic system, additional to drainage and surface runoff.*

We will change this passage to the following wording and cite Abril and Borges (2019) here.

"Therefore, these deltas experience seasonal flooding, instead of (semi)-diurnal flooding determined by tidal action. Flooding can, in addition to groundwater drainage and surface runoff, transport substantial amounts of terrestrial carbon to aquatic systems (Abril & Borges, 2019). We thus anticipate seasonal variability in $CO_2$ and $CH_4$ emissions and in lateral carbon transport from the Danube Delta to the ocean. "

*L162: "As tests showed that there was no significant difference between the lab- and field-based methods, we pooled the data in our analysis." I suggest you provide the result of these tests as a figure as supplementary material*

We will add the following paragraph to the supplementary material, section 1:

"In October 2017, we conducted a comparison of $CH_4$ measurement procedures using the GC and the Los Gatos using field samples from the Danube Delta. We calculated average values for the lab-based GC procedure (n=2) and the field-based LG procedure (n=3). Considering the standard deviation of the samples, only 2 samples deviate from the 1:1 line, however they are still within the 10% measurement uncertainty of our GC system. Based on the results of this comparison, we deemed it appropriate to pool data acquired using the two different methods."

[Figure]

*Figure 1 Average CH$_4$ concentration measured with lab-based GC method (n=2) versus field-based LG method (n=3). Error bars show the standard deviation, the orange line symbolizes the 1:1 line.*

*L184: "In the high-resolution LGR time series, the influence of gas bubbles could easily be identified." The method is described graphically in Grasset et al. Freshwater Biology. doi:10.1111/fwb.12780.*

Thank you for the reference. Grasset et al. (2016), however, calculated the total flux as the diffusive flux determined from a linear regression plus the partial pressure increase during an ebullition event divided by the total observation time, i.e. $F_{tot} = F_{dif} + \frac{\Delta p_{ebu}}{t}$. In our study, we calculated the total flux as the difference between the initial and the final observed CH4 partial pressure, i.e. $F_{tot} = \frac{p_{final} - p_{initial}}{t}$, a method that was for example also used by Beaulieu et al. (2016).

A visual representation of our approach is provided in the supplementary material and we will add a reference to Beaulieu et al. (2016).

*L242 In section "2.6 Import by Danube River and Export to Black Sea", please provide here or maybe in the results section, more precise verbal information (equation is ok) on how you calculate C lateral fluxes before and after the wetland in the delta and **how you deal with the problem that these two fluxes might be too close to each other to allow a precise calculation of the net lateral export from the wetland in the delta as a small differences between two large numbers that contain some uncertainty**. What are **the representativity of stations and data**, with respect to observed spatial and temporal differences in the C forms and discharge data between sampling points.*

"We calculated the lateral transfer of carbon between the Danube Delta and its River by subtracting the load exported to the Black Sea via the three main branches from the load imported to the Delta from the catchment:

$$F_{lateral} = F_{St1} - (F_{St3} + F_{St4} + F_{St5})$$

Station 1 is located in the Tulcea branch close to the apex of the delta and represents the water signature from the catchment, while stations 3, 4 and 5 are located in the 3 main branches close to the Black Sea. Stations 4 and 5 are located shortly upstream of the settlements of Sulina and St.George to avoid measuring the effect of these two settlements. Station 3 is located in a small side arm of the Chilia branch marking the border between Romania and Ukraine, which during comparison measurements showed the same water composition as the main branch.

The resulting lateral flux in our case is comparably small and we used gaussian error propagation to estimate its range. The basis were the measurement uncertainties in concentrations (0.5% DIC, 4% DOC, 10% POC) and discharge (3%, assumed), which were used to calculate the loads".

*L290: figure 3 and next ones would be easier to read if simultaneous discharge could be shown*

We agree and will add the discharge (daily average discharge close to the apex at Isaccea, also shown in Fig 2) as additional panel above right figure panel of Fig 4 (see below).

*L350 and k values. Lake are more exposed to wind indeed, however rivers and channels are exposed to current which may also contribute to $k_{600}$*

We are of the opinion, that the display of $k_{600}$ is most meaningful with respect to the three different categories and therefore add this plot next to the seasonal plot of the discharge in Fig 4. See below for the upper section of the updated Fig. 4.

[Figure]

Figure 2 Figure 4 $k_{600}$ (a), daily average discharge close to the apex (b) and measured concentrations of dissolved gases in the different waterscapes, i.e. river, channel and lake (a, c-h). Left panels (a, c, e, g): pooled data from 2-years. Right panel (d, f, h): seasonal dynamics with dotted lines connecting median values. X-axis ticks indicate day 15 of the respective month. c) & d) CH4 in 2016: four channel values (ranging from 22.2 to 58.0 µM) and one lake station (12.5 µM) exceeding 10 µM were cutoff. e) & f) dotted black line represents equilibrium concentration of CO2 at 15°C (18.2 µM). Boxplots indicate 25 and 75 percentiles, as well as median, whiskers indicate maximum and minimum, with data > 1.5*IQR is shown as outliers

*L380 you are repeating what has been said in Mat and met about $CH_4$*

Thank you, we will delete the respective sentence here.

*L394. The calculation of lateral flux is indeed poorly constrained and it would be interesting to see the data that support the statements "POC import from the catchment exceeds the export to the Black Sea in February and March, while DOC import exceeds export only during August (data not shown)."*

We will add this data in the form of a Figure to the appendix and rephrase the paragraph as following:

"The water export from the delta, however, is poorly constrained. The balance between precipitation minus evaporation is negative, poorly quantified and quite variable. We therefore rely on the flux balance of the three branches to estimate carbon export from the delta. The resulting export to the Black Sea via the Danube's main branches amounts to 8650 ± 147 GgC yr$^{-1}$ and is less than 2 % higher than the inflow load reaching the apex of delta. It mainly relates to increased DOC levels reaching the main branches from the delta, especially during the spring flood in March and April. The relatively small fraction of water that passes through the delta changes the relative fraction of DOC and POC only marginally to 7 % and 4 %, respectively, while

the largest fraction in the water reaching the Black Sea remains DIC (89 %, Fig. 8). DIC import and export is fairly comparable throughout the year, while POC export to the Black Sea strongly exceeded the imports from the catchment in April. DOC exports are highest in the first half of the year (see Fig Sxy)."

Supplementary Information:

"3.x Seasonality of carbon import and export

Import and Export loads varied seasonally, which was to a large extent driven by variations in discharge (Fig. Sxy).

[Figure]

Figure Sxy: Carbon import to the delta and export to the Black Sea (sum of Chilia, Sulina and St. George branch) in the forms of DIC, DOC and POC loads. Please observe the different order of magnitude for DIC."

*L400-410, please simplify the story about statistics.*
The simplified version will read as follows:

"The non-parametric Kruskal-Wallis test does not require normal distribution of the data, but it requires equal variance of the data groups investigated for difference in median (Hedderich & Sachs, 2016). Our observations in the seasonal plots (Fig. 3 & 4) support the results of the test: in most cases, the boxplots do not overlap, indicating that the three groups are significantly different. For example, DOC is significantly higher in the delta lakes and channels due to the strong primary productivity of these systems. $O_2$ is significantly lower in the channels than in the other two categories due to lateral inflow of oxygen-depleted waters from the wetland (Zuijdgeest et al., 2016; Zurbrügg et al., 2012). The large difference between the waterscapes with respect to $CO_2$ and $CH_4$ fluxes supports our approach to treat the waterscapes independently when upscaling the flux measurements to the total water surface of the delta."

Unfortunately, we did not measure depth profiles and I couldn't find anything on summer stratification in Danube Delta lakes. The shallow character of most investigated lakes (maximum depth of 3.5 m in Lake Rosu) led us to speculate that the lakes may be fully mixed or maybe stratified during individual days in summer. Since we don't have any prove for either condition, we did not speculate about it at this point.

*L445: the study of $CH_4$ emissions from various plant types by Grasset et al 2016 might be helpful here*

Thank you for the reference. We plan to adapt this paragraph as follows:

"In the lakes, longer residence times of 10–30 days allow primary production and local decomposition of organic matter to become important factors driving carbon cycling. We observed abundant macrophytes like *Ceratophyllum demersum* and *Elodea canadensis* growing in spring and early summer, which, depending on lake depth, even reached the lake water surface. A change in abundance of submerged vegetation to vegetation with floating leaves might be linked to changes in the $CO_2$ and $CH_4$ fluxes (Grasset et al., 2016). Around July, algal blooms coincided with a significant reduction in macrophyte abundance. This pattern seems to be reoccurring due to the eutrophic state of the delta lakes (Coops et al., 2008; Coops et al., 1999; Tudorancea & Tudorancea, 2006). During our observations, both macrophytes and algal blooms caused a drawdown of $CO_2$ and supersaturation in $O_2$ (Fig. 4d & 4f). The algal blooms also partly explain the peak in measured POC from July to November, which extended to most of the delta's channels (Fig. 3d). The degradation of the macrophyte biomass coincided with locally elevated $CH_4$ concentrations from July to October (Fig. 4b).

*L584 drylands are not defined as the contrary of wetlands. The difference here is between floodable land and non-floodable land*

Yes, thank you. We will correct this:

"Abril and Borges (2019) recently suggested that the active pipe concept of carbon transport in the aquatic continuum indeed needs to be extended to consider floodable and non-floodable land as separate carbon sources."

*L594 "Assuming a carbon content of 0.42 gC gBiomass-1 determined by Greenway and Woolley (1999) for Phragmites australis, between 1000 and 1210 GgC yr$^{-1}$ are bound in the form of macrophyte biomass in the reed area of the Danube Delta". Not clear the meaning of "bound" here. A great portion of the macrophyte biomass is supposed to be recycled. "The wetlands thus hold 12 to 17 times the total input of organic C to the delta from the catchment" do you mean the total ANNUAL input?*

The idea was to compare input of organic carbon from the catchment with the primary production from the vegetation. The annual primary production of the reed is between 1000-1210 GgC yr$^{-1}$, while we estimate the total annual input of organic carbon from the catchment to 79 GgC yr$^{-1}$ (i.e. 10% of the organic carbon load transported by the river). You are right, the term "bound" is ambiguous in this context, since we are referring to the reed biomass and not the carbon stored in the sediment. Please see comment L600

*L595 "Nevertheless, wetlands are considered to be net C sinks and Zhou et al. (2009) estimate the sink capacity of a Phragmites australis dominated wetland to -62 g C m$^{-2}$ yr$^{-1}$ considering $CO_2$ and $CH_4$ release from the wetland itself." You must be more precise here in the vocabulary used. $CO_2$ sink might be different from C sink. Please specify that C sink is OC burial in sediments and not the atmospheric $CO_2$ sink. Same in the following sentences.*

Thank you, we will correct the imprecise wording, see comment L600.

*L600 "Assuming that carbon emitted from the channels originated only from the wetland source, this would suggest that up to 20 % of the potential wetland sink might be exported laterally, eventually finding its way to the atmosphere." Please explain more clearly*

Since the comments L594, L595 and L600 all concern the same paragraph and are somehow linked to each other, we present the planned changes as follows:

"The Danube Delta is dominated by the plant association Scirpo-Phragmitetum, which covers nearly 89% of the total marsh area (1600 km$^2$). Its net primary productivity ranges between 1500–1800 g m$^{-2}$ yr$^{-1}$ (Sarbu, 2006), which is slightly higher than the average net primary productivity of intertidal salt marshes and mangroves (1275 gC m$^{-2}$ yr$^{-1}$) (Cai, 2011). Assuming a carbon content of 0.42 gC gBiomass$^{-1}$ determined by Greenway and Woolley (1999) for *Phragmites australis*, primary production in the reed amounts to 1000–1210 GgC yr$^{-1}$ (Fig. 8), which is about 8 times less than the carbon load transported by the river. A large fraction of the net carbon assimilation by the *phragmites* stands is decomposed and released back to the atmosphere. In a Danish wetland, more than 50 % of the carbon was respired and released back to the atmosphere, with 48 % being released as $CO_2$ and 4 % as $CH_4$ (Brix et al., 2001). In the Danube Delta, the 50 % accretion rate would correspond to about 500 gC m$^{-2}$ yr$^{-1}$. However, net primary production and carbon accretion change seasonally with environmental factors such as temperature and irradiation. Accordingly, net $CO_2$ assimilation in the Danish study was limited to the warm season from Arpil to September, whereas $CO_2$ and $CH_4$ emission occurred during the whole year but with maxima of 0.2 mol Cm$^{-2}$ d$^{-1}$ during July-August. Qualitatively, we observed the same seasonality in $CO_2$ oversaturation in the Channels that drain water from the *Phragmites* stands (Fig. 4d).

For a *phragmites australis* dominated wetland in China, at a latitude comparable to the Danube Delta, Zhou et al. (2009) estimated the annual net uptake of $CO_2$ to 62 gC m$^{-2}$ yr$^{-1}$. Scaled to the area of the Danube Delta, this would result in 99 GgC yr$^{-1}$ remaining in the delta, which is in the same order of magnitude as the total annual input of organic C from the catchment (79 GgC yr$^{-1}$). Similar to the Danish study, also Zhou et al. (2009) do not account for potential lateral transport of carbon to adjacent water bodies. Our results show that channels in the Danube Delta are receiving carbon from the wetland, with peaks in $CO_2$ and $CH_4$ concentrations that match the maxima in the gross ecosystem production in China. Comparing the estimated carbon fluxes from the channels with the yearly carbon accumulation estimates of the wetland suggests that up to 20% of the latter could be released to the atmosphere via lateral transport, assuming the carbon flux from the channel were exclusively sustained by the wetland. With a lag phase of about 3 months, the Danube Delta reed beds release peak concentrations of DOC and POC during October to November, when the biomass in the reed stands start degrading (Figure 3 d,e)."

References:

Abril, G., & Borges, A. V. (2019). Ideas and perspectives: Carbon leaks from flooded land: do we need to replumb the inland water active pipe? *Biogeosciences, 16*(3), 769-784. doi:10.5194/bg-16-769-2019

Aufdenkampe, A. K., Mayorga, E., Raymond, P. A., Melack, J. M., Doney, S. C., Alin, S. R., . . . Yoo, K. (2011). Riverine coupling of biogeochemical cycles between land, oceans, and atmosphere. *Frontiers in Ecology and the Environment, 9*(1), 53-60.

Beaulieu, J. J., McManus, M. G., & Nietch, C. T. (2016). Estimates of reservoir methane emissions based on a spatially balanced probabilistic-survey. *LIMNOLOGY AND OCEANOGRAPHY, 61*(S1), S27-S40. doi:10.1002/lno.10284

Begy, R. C., Simon, H., Kelemen, S., & Preoteasa, L. (2018). Investigation of sedimentation rates and sediment dynamics in Danube Delta lake system (Romania) by 210Pb dating method. *Journal of Environmental Radioactivity, 192*, 95-104. doi:https://doi.org/10.1016/j.jenvrad.2018.06.010

Brix, H., Sorrell, B. K., & Lorenzen, B. (2001). Are Phragmites-dominated wetlands a net source or net sink of greenhouse gases? *Aquatic Botany, 69*(2), 313-324. doi:https://doi.org/10.1016/S0304-3770(01)00145-0

Cai, W.-J. (2011). Estuarine and Coastal Ocean Carbon Paradox: CO2 Sinks or Sites of Terrestrial Carbon Incineration? *Annual Review of Marine Science, 3*(1), 123-145. doi:10.1146/annurev-marine-120709-142723

Cole, J. J., Prairie, Y. T., Caraco, N. F., McDowell, W. H., Tranvik, L. J., Striegl, R. G., . . . Melack, J. (2007). Plumbing the global carbon cycle: Integrating inland waters into the terrestrial carbon budget. *Ecosystems, 10*(1), 171-184. doi:10.1007/s10021-006-9013-8

Coops, H., Buijse, L. L., Buijse, A. D. T., Constantinescu, A., Covaliov, S., Hanganu, J., . . . Oosterberg, W. (2008). Trophic gradients in a large-river Delta: ecological structure determined by connectivity gradients in the Danube Delta (Romania). *River Research and Applications, 24*(5), 698-709.

Coops, H., Hanganu, J., Tudor, M., & Oosterberg, W. (1999). Classification of Danube Delta lakes based on aquatic vegetation and turbidity *Biology, Ecology and Management of Aquatic Plants* (pp. 187-191): Springer.

Grasset, C., Abril, G., Guillard, L., Delolme, C., & Bornette, G. (2016). Carbon emission along a eutrophication gradient in temperate riverine wetlands: effect of primary productivity and plant community composition. *Freshwater Biology, 61*(9), 1405-1420. doi:10.1111/fwb.12780

Greenway, M., & Woolley, A. (1999). Constructed wetlands in Queensland: Performance efficiency and nutrient bioaccumulation. *Ecological Engineering, 12*(1), 39-55. doi:https://doi.org/10.1016/S0925-8574(98)00053-6

Hastie, A., Lauerwald, R., Ciais, P., & Regnier, P. (2019). Aquatic carbon fluxes dampen the overall variation of net ecosystem productivity in the Amazon basin: An analysis of the interannual variability in the boundless carbon cycle. *Global Change Biology, 25*(6), 2094-2111. doi:10.1111/gcb.14620

Hedderich, J., & Sachs, L. (2016). *Angewandte Statistik*: Springer.

Sarbu, A. (2006). Aquatic macrophytes. In C. Tudorancea & M. M. Tudorancea (Eds.), *Danube Delta: genesis and biodiversity* (pp. 133-175). Leiden: Backhuys Publishers.

Stumm, W., & Morgan, J. (1981). Aquatic chemistry 2nd edition: John Wiley & sons.

Tootchi, A., Jost, A., & Ducharne, A. (2019). Multi-source global wetland maps combining surface water imagery and groundwater constraints. *Earth System Science Data, 11*, 189 - 220. doi:10.5194/essd-11-189-2019

Tudorancea, C., & Tudorancea, M. M. (2006). *Danube Delta: genesis and biodiversity*. Leiden: Backhuys Publishers.

Zhou, L., Zhou, G., & Jia, Q. (2009). Annual cycle of CO2 exchange over a reed (Phragmites australis) wetland in Northeast China. *Aquatic Botany, 91*(2), 91-98. doi:https://doi.org/10.1016/j.aquabot.2009.03.002

Zuijdgeest, A., Baumgartner, S., & Wehrli, B. (2016). Hysteresis effects in organic matter turnover in a tropical floodplain during a flood cycle. *Biogeochemistry, 131*(1), 49-63. doi:10.1007/s10533-016-0263-z

Zurbrügg, R., Wamulume, J., Kamanga, R., Wehrli, B., & Senn, D. B. (2012). River-floodplain exchange and its effects on the fluvial oxygen regime in a large tropical river system (Kafue Flats, Zambia). *Journal of Geophysical Research: Biogeosciences, 117*(G3). doi:doi:10.1029/2011JG001853